# Microencapsulation of Essential Oils: A Review

**DOI:** 10.3390/polym14091730

**Published:** 2022-04-23

**Authors:** Vânia Isabel Sousa, Joana Filipa Parente, Juliana Filipa Marques, Marta Adriana Forte, Carlos José Tavares

**Affiliations:** Physics Center of Minho and Porto Universities (CF-UM-PT), Campus of Azurém, University of Minho, 4804-533 Guimarães, Portugal; vaniafernandesousa@gmail.com (V.I.S.); joanacp_17@hotmail.com (J.F.P.); juliana.g.marques@hotmail.com (J.F.M.); martadrianaff@gmail.com (M.A.F.)

**Keywords:** essential oils, extraction techniques, microencapsulation, controlled release, microcapsules, pharmacology

## Abstract

Essential oils (EOs) are complex mixtures of volatile compounds extracted from different parts of plants by different methods. There is a large diversity of these natural substances with varying properties that lead to their common use in several areas. The agrochemical, pharmaceutical, medical, food, and textile industry, as well as cosmetic and hygiene applications are some of the areas where EOs are widely included. To overcome the limitation of EOs being highly volatile and reactive, microencapsulation has become one of the preferred methods to retain and control these compounds. This review explores the techniques for extracting essential oils from aromatic plant matter. Microencapsulation strategies and the available technologies are also reviewed, along with an in-depth overview of the current research and application of microencapsulated EOs.

## 1. Introduction

Essential oils (EOs) are liquid products present in plants and can be defined as complex natural mixtures of volatile secondary lipophilic metabolites that give plants and spices their essence and colour [1,2]. These compounds can be obtained by hydrodistillation, solvent extraction, and supercritical CO_2_ extraction, among other methods that can also be used for essential oil extraction [1]. These oils can be extracted from different parts of the plant, such as the flowers, leaves, stems, roots, fruits, and bark, and have different biological and pharmaceutical properties [3]. Due to their versatile nature, the oils can be utilised for several purposes, from contact toxicant and fumigant to attractive or repellent applications [4]. However, there are many factors that affect the chemical composition of essential oils, including genetic variation, type or variety of plants, plant nutrition, fertilizer applications, geographical location of the plant, climate, seasonal variations, stress during growth or maturation, as well as post-harvest drying and storage.

EOs that have antimicrobial properties are alternatives to the use of antibiotics and chemical additives [5]. As they have been used worldwide in many industries, their prices differ due to the supply of raw materials, issues related to harvesting, climate factors, and extraction yields. Some of these EOs also have antioxidant properties, with studies reporting that EOs from celery, citronella, cloves, oregano, parsley, tarragon, and thyme seeds were able to inhibit 50% of the 2,2-diphenylpicliryl-hydrazil (DPPH) radical elimination activity [6]. The application of EOs as antioxidants has been evaluated in different types of foods, and research is currently being conducted to optimise the process [7].

Essential oils have gained renewed interest in several areas over the years. Its use was expanded to the medical field due to its biocidal activities (bactericides, viricides, and fungicides) and medicinal properties [8]. The use of natural compounds has become popular in the food industry, with EOs being used as preservatives and food additives due to their antioxidant and antimicrobial properties and pleasant flavour. EOs are included in the composition of many dosage forms in pharmaceutical products. Studies have been carried out on the many biological activities of essential oils (Figure 1) and their components, and particular interests have also been established to elucidate their modes of action, allowing for the improved and targeted intervention in new drugs [9].

EOs are unstable and highly susceptible to changes caused by external factors, such as light, temperature, oxygen, and humidity [10]. The high volatility and reactivity of these compounds represent challenges for the application of essential oils in several industries [11]. To overcome these limitations, the microencapsulation technique is often used to maintain the functional and biological characteristics of these compounds and to control their release [12]. 

Microencapsulation is a technology based on the coating of solid, liquid, or gaseous particles through an encapsulating agent that acts as a barrier, completely isolating the core material from the external environment [13]. Most microcapsules have a diameter within the range 1–1000 µm [14]. The shell material can be a film of a natural, semi-synthetic, or synthetic polymer and its choice has a key role in the stability of core material [15]. Arabic gum, agar, alginate, proteins, and dextrins are some of the materials used as encapsulating agents in the microencapsulation process [16]. 

Due to the enormous interest of the scientific community and the industry in the microencapsulation of active substances, several microencapsulation methods have been developed over time. Encapsulation processes are usually divided into three main categories: physicochemical, mechanical, and chemical processes [17]. In this review article, some of these methods, which are used in EO microencapsulation, are described and an outlook of scientific works developed in this area is approached.

## 2. Essential Oils

Essential oils are defined, according to the *European Pharmacopoeia,* as an “odorous product, usually of complex composition, obtained from a botanically defined plant raw material by steam distillation, dry distillation, or a suitable mechanical process without heating. EOs are usually separated from the aqueous phase by a physical process that does not significantly affect their composition” [18]. EOs are extracted from aromatic plant materials such as oily aromatic liquids, and they can be biosynthesised in different plant organs as secondary metabolites, such as flowers, herbs, buds, leaves, fruits, branches, bark, zest, seeds, wood, rhizomes, and roots.

Essential oils are complex mixtures of highly volatile aromatic compounds named after the plant from which they are derived. Within the different species of plants, only 10% contain EOs and are called aromatic plants. These natural products exert the function of protecting the plants, guaranteeing the growth of the plant and the propagation of species. Essential oil provides the essence, odour, or flavour of the plant and some of the functions that it performs in plants can also be made in living organisms [19]. EOs are generally liquid at room temperature and are hydrophobic (immiscible with water) and lipophilic (miscible with other oils and organic solvents) substances [20]. In general, essential oils are a mixture of compounds with their own physicochemical characteristics that, when combined, give the oil a particular odour. The different aroma of oils is fundamentally due to variations in the volatility and relative concentration of its constituents [21].

### 2.1. Chemistry of Essential Oils

The chemical composition of EOs can be complex due to the number of different components, which can have promising chemical and biological properties [22]. Essential oils are complex mixtures that can contain over 300 different compounds. Most EOs are characterised by two or three main components in reasonably high concentrations (20–70%) compared to other components present in small amounts [8]. The organic constituents have a low molecular weight, and their vapour pressure (at atmospheric pressure and at room temperature) is high enough for them to be partially in vapour state [23].

Chemically, EOs mainly belong to two classes of compounds: terpenes and phenylpropanoids (Table 1). The terpene family is predominant, and phenylpropanoids, when they appear, are responsible for the characteristic odour and taste [24].

#### 2.1.1. Terpenoids

Terpenes, also called terpenoids, constitute the largest class of natural products with several structurally diversified known compounds [25]. Their structures contain carbon skeletons and are formed by isoprene units, being classified according to the number of these units that compose their structure. They can be classified as hemiterpenes (1 isoprene unit; 5 carbons), monoterpenes (2 isoprene units; 10 carbons), sesquiterpenes (3 isoprene units; 15 carbons), diterpenes (4 isoprene units; 20 carbons), triterpenes (6 isoprene units; 30 carbons), and tetraterpenes (8 isoprene units; 40 carbons), among others. Monoterpenes and sesquiterpenes are mostly found in volatile essential oils. Terpenes can present aromatic, aliphatic, and cyclic structures and can contain oxygen atoms, being called terpenoids (Figure 2) [26].

#### 2.1.2. Phenylpropanoids

Phenylpropanoids are natural substances commonly found in plants and consist of a six-carbon aromatic ring joined to a three-carbon side chain. This side chain contains a double bond and the aromatic ring may be substituted. These compounds are biosynthesised from shikimic acid, which forms the basic units of cinnamic and *p*-coumaric acids. These units, through enzymatic reductions, produce propenylbenzenes and/or allylbenzenes and, through oxidations with side chain degradation, generate aromatic aldehydes [27,28].

**Table 1 polymers-14-01730-t001:** Composition of compounds found in essential oils [29].

Essential Oil Compounds		
Classes		Constituents
Terpenes	Monoterpene	(−)-Camphene, p-cymene, (+)-limonene, β-ocimene α-phellandrene, α-pinene, α-terpinene, terpinoleneorange,
	Sesquiterpene	(−)-β-isabolene, α-cadinene, β-caryophyllene, α-copaene, β-elemene, α-farnesene, α-humulene, α-zingiberene
Phenylpropanoids		(E)-Anethole, cinnamaldehyde, cinnamic acid cinnamic alcohol, eugenol, methyleugenol, myristicin

### 2.2. Extraction Methods

Aromatic herbs or parts thereof, such as leaves, flowers, bark, seeds, and fruits, are subjected to extraction processes after being collected at specific stages of maturity and stored under controlled conditions (light, temperature, and humidity). 

Extraction techniques are essentially divided into classical and conventional methods and innovative methods. Classical methods are based on the distillation of water by heating, to extract the EOs from the plant matter. Hydrodistillation, steam distillation, hydrodiffusion, organic solvent extraction, and cold pressing are some of these methods. New extraction technologies have been developed in order to overcome some of the disadvantages of conventional methods. Methods such as ultrasound-assisted extraction and microwave-assisted extraction use energy sources that make the process more environmentally friendly. On the other hand, methods such as supercritical fluid extraction and subcritical liquid extraction allow the non-polar components from the material to be extracted [30].

#### 2.2.1. Hydrodistillation

Hydrodistillation is the oldest and simplest method for extracting OEs. This method is characterised by direct contact between the solvent and the plant material, that is, the raw material is submerged in boiling water (Figure 3) [31]. 

In this procedure, the cell walls are broken, and the oil is evaporated together with the water, and then condensed into a mixture of water vapour and volatile compounds of vegetable raw material. However, these two phases (volatile compounds and water) are immiscible, rendering possible an additional separation according to the difference in density [32]. This technique is inexpensive, but, at the same time, it is not selective because of the waste of large amounts of the compound in the solvent (part of the extract can be lost in the aqueous phase) and can provide low yields [33,34]. Despite being the oldest method, hydrodistillation is still used today for extracting oils from different matrices. Essential oils from *Rosmarinus officinalis* L. [35], *Ziziphora clinopodioides* L. [36], *Citrus latifolia Tanaka* [37], and *Zingiber officinale* [38] are some of the medicinal plants where EOs can be extracted by hydrodistillation.

#### 2.2.2. Steam Distillation

Steam distillation is one of the preferred methods of extracting EOs. The extraction procedure is based on the same principles as hydrodistillation. The difference essentially lies in the absence of contact between the substrate to be extracted and the water, which causes a reduction in the extraction time.

The sample is placed in a column where the bottom part is connected to a flask with water under heating (Figure 4). The top part is connected to a condenser, where the steam produced passes through the sample, taking essential oils to the condenser. This process causes the condensation of the water–oil mixture, and this mixture can be separated by liquid–liquid extraction [39].

This method is applied commercially and on a large scale in the extraction of essential oils from hops [40] and in the extraction of several EOs such as lavender [41] and patchouli essential oil [42].

#### 2.2.3. Organic Solvent Extraction

Some essential oils (such as rose and jasmine) have low thermal stability and are unable to withstand high temperatures. In these cases, organic solvents that have a low boiling temperature, are chemically inert, and have low cost can be used.

In organic solvent extraction, the sample is placed in contact with the organic solvent (which can be hexane, benzene, toluene, or petroleum ether, among others) for a period that allows the transfer of the soluble content of the sample. The extracted matrix is concentrated by evaporating the solvent present in the liquid phase. This method allows the sample to be permanently in contact with a quantity of fresh solvent and, at the end of the process, it is not necessary to carry out filtration, as long as there are high yields [34]. Solvent extraction is the most-used conventional method in the cosmetic industry [43,44,45]. Figure 5 represents the extraction of organic solvents through a Soxhlet extraction [32,46].

#### 2.2.4. Cold Pressing

Essential oils are mechanically removed by cold pressing, where the oil glands are broken and volatile oils are released. In this process, an aqueous emulsion is formed, where the oil present can be obtained through centrifugation, decantation, or fractional distillation [13]. The cold pressing method is essentially used to extract oils from citrus fruits [47,48,49]. 

#### 2.2.5. Supercritical Fluid Extraction (SCFE)

Supercritical fluid extraction is an efficient, environmentally friendly, and clean technique for isolating EOs. In this technique, supercritical fluids are used as extraction agents due to the supercritical state of fluids, conferring excellent characteristics for the extraction process, such as low viscosity, high density (close to that of a liquid), and high diffusivity (high penetration power).

Several substances can be used as supercritical solvents, such as water, carbon dioxide (CO_2_), methane, ethylene, and ethane. However, CO_2_ is the most-used solvent due to its critical point being easily reached (low temperature and pressure, 31.2 °C and 72.9 atm, respectively), low toxicity and reactivity, low cost, and non-flammability. After selecting the ideal temperature and pressure for extraction, supercritical fluid passes through the sample and the oils are dissolved and extracted. Subsequently, the extraction solution is maintained at a pressure below the critical point and as the pressure decreases, the supercritical fluid passes to the gaseous state and loses its solvating capacity, being recycled [50]. This method is increasingly used commercially, being applied in the extraction of EOs from the leaves of laurel [51], rosemary [52], sage [53], flowering plants [54], and horseradish tree [55].

#### 2.2.6. Microwave-Assisted Extraction (MAE) 

Due to the need to use more ecological and energy-efficient extraction methods, microwave-assisted extraction has become an alternative to conventional methods. The sample is placed in a microwave reactor without any solvent, where the electromagnetic energy that is converted into heat increases the internal temperature of sample cells due to the evaporation of the moisture present. The internal pressure increases, the glands rupture, and the essential oil is released [56]. Several EOs were extracted from plant matrices through this technique, such as orange [57], laurel [58], lemon [59], mint [60], rosemary [61], and basil [62]. 

#### 2.2.7. Ultrasound-Assisted Extraction (UAE) 

Ultrasound energy allows the intensification of EO extraction [63]. Therefore, it is usually combined with other extraction techniques in order to accelerate the extraction process and increase the speed of mass transfer. The sample is submerged in a solvent while being subjected to ultrasound. This method, through rapid solvent movements, induces a mechanical vibration of the walls and membranes of the sample that causes the release of essential oils. In some areas, it is already considered a large-scale application method, such as in the medical and food industry, where it is used to increase the quality of the extracted substrate, reduce working time, and increase the yield [64]. 

In general, alternative methods have emerged to overcome some of the disadvantages and limitations of conventional methods. Traditional methods have long extraction times (4 to 6 h), high energy consumption, and use solvents that increase environmental pollution. Furthermore, they can cause chemical changes to the EOs that are thermally unstable, causing a decrease in the quality of the extracted oils and changes in the chemical nature of compounds. The ‘greener’ alternatives are more sustainable and economical due to reduced water and energy consumption and reduced CO_2_ emissions. However, these methods are not easily accessible, and the initial investment is higher. 

Therefore, currently, the hydrodistillation method continues to be the most-used extraction technique in laboratory due to its accessibility, simplicity, and lower cost [46]. Table 2 summarises the advantages and disadvantages of the different EO extraction methods.

### 2.3. Essential Oils Application

#### 2.3.1. Essential Oils in Plants 

EOs are stored in specific parts of plants, acting in extraordinarily different ways. Some aromatic plants have been widely explored due to their properties, such as bay laurel (*Laurus nobilis*). This plant is an aromatic tree, and laurel oil is extracted from the dry leaves and branches, appearing as a greenish yellow liquid with a powerful medicinal odour. In addition to being used in cuisine, the laurel tree leaves are used in medicine for having antioxidant [65], antibacterial [65,66], and antifungal [22] properties. According to the literature, laurel has also been proven to be an insect repellent [67,68]. However, it can cause dermatitis in some individuals, and due to the possible narcotic properties attributed to methyleugenol, this oil should be used in moderation.

*Cymbopogon nardus*, commonly known as citronella, is an aromatic and perennial herb. Citronella oil can also be produced from Java or Maha Pengiri citronella (*C. winterianus*) [69]. Citronella leaves are used for their aromatic and medicinal value in many cultures, such as in the treatment of fever, intestinal parasites, and digestive and menstrual problems, as well as for use as an insect stimulant and repellent [69,70,71,72]. Citronella is also used in traditional Chinese medicine for rheumatic pain, and it has antifungal [73], antioxidant, and antibacterial [74] properties. It is non-toxic and non-irritating, but it can cause dermatitis in some people [69].

Regarding the medical properties of hops (*Humulus lupulus*), these are better known for treatments associated with nerves, insomnia, nervous tension, neuralgia, and for sexual neurosis in both sexes [5]. It has antibacterial [75,76], antifungal [76], anti-cancer [76,77], and repellent [78,79] properties. In China, it is used for pulmonary tuberculosis and cystitis treatment. It can also be used to make beer. It is non-toxic and non-irritating, but it can cause sensitivity in some individuals, and people with depression should avoid this oil [76]. 

Lemon balm (*Melissa officinalis*) is a herbaceous perennial from the mint family and it has antibacterial, antifungal [80], sedative, antipyretic, antispasmodic, anti-hypertensive, anti-Alzheimer, and antiseptic properties [81]. In addition to the treatment of several gastrointestinal, liver, and nervous system disorders, it has also been reported that lemon balm is useful in the treatment of asthma, bronchitis, coughs, and several pains [82]. Furthermore, this plant is notably marked by its antimicrobial applications in different medicines, exemplified by its use in insect bites (wasps and bees) and poisonous or infectious bites [81,83]. 

*Azadirachta indica*, better known as neem, is an ancient tree that has been used for centuries for the most varied purposes. The plant provides a great number of secondary metabolites with biological activity. The plant has gained great importance in several areas, such as agriculture, livestock, and medicine [84]. It is used as an insecticide [85], antiviral [86], antibacterial [87], and antimicrobial [88], among others. Neem oil is very effective for acne, psoriasis, and eczema treatments, but it can also be applied as a support in the treatment of topical fungal or viral conditions, such as nail fungus, athlete’s foot, warts, or wounds. The natural antihistamines contained in neem oil are effective in relieving the itching and burning caused by, for example, bee, mosquito, and spider bites. The main constituent of neem is azadiractin, found in the leaves, fruits, and seeds. 

*Mentha pulegium*, better known as pennyroyal or mint (Brazil), is one of the best-known species of the genus *Mentha*. Pennyroyal extracts are good insect repellents [89,90,91,92]. There are several studies that show that these extracts also have other properties, such as antimicrobial [92,93,94], antioxidant [92,93,95], antibacterial [95,96], and anti-tumour [96] uses. It is still current in the *British Herbal Pharmacopoeia*, indicated for flatulent dyspepsia, intestinal colic, common cold, delayed menstruation, skin rashes, and gout [69]. 

*Illicium verum*, popularly known as star anise, is a plant considered a spice for medicinal and culinary use. The extraction of *Illicium verum* has carminative, stomach, stimulating, and diuretic properties and is used as a pharmaceutical supplement [97]. The extracted shikimic acid is one of the main ingredients of the antiviral drug Tamiflu^®^ (oseltamivir) that is used to treat symptoms caused by avian influenza [98]. It has also been reported to have antimicrobial properties [99] and antioxidant properties [100], as well as significant anti-cancer potential [101]. There are studies in which star anise has been used as an insect repellent, such as for the Indian flour moth (*P. interpunctella larvae*) [102,103]. The main constituent of star anise is trans-anethole (Table 3) (80–90%) and when used in large doses, it is narcotic and slows down circulation, which can lead to brain disorders [69].

Valerian is a perennial flowering plant with many chemical constituents, with more than 150 constituents identified in its essential oil [104]. Regarding therapeutic indications, it is advisable for people with nervous agitation, mild anxiety, and difficulties in sleeping [105,106]. There are also studies that demonstrate its antibacterial [107] and antimicrobial [108] properties. It is non-toxic, non-irritating, and can cause sensitisation. Table 3 describes the main components of the EOs present in plants, as well as their chemical structures and some of the biological properties.

#### 2.3.2. Pharmacological and Medical Applications

Essential oils have a wide range of biological properties, and there has been a growing interest in clinical applications (Table 4). Some of the properties include the chemo-preventive effects of cancer [109], antifungal [110], antiviral [111], antimicrobial, analgesic, anti-inflammatory [112], and antiparasitic activities [113].

An extensive range of EOs have antibacterial activity against Gram-positive and Gram-negative bacteria, along with antifungal properties. These compounds have been studied and have shown very promising results in salmonella, staphylococci, and other oral pathogens, and can be an alternative to antibiotics providing they are properly studied for these effects [115,116]. EOs that have shown antibacterial potential are basil [117], manuka oil (more potent among eucalyptus oil, rosmarinus, lavandula, and tea tree) [118], melaleuca oil [119], and essential leaf oils *P. undulatum* and *Hedychium gardnerianum*.

With regard to antifungal activity, melaleuca oil showed positive results for all of its constituents, especially against dermatophytes and filamentous fungi [120]. In a reported study, germinated conidia of *Aspergillus niger* were more susceptible to non-germinated ones, with EOs of *Melaleuca ericifolia*, *Melaleuca armillaris*, *Melaleuca leucadendron*, and *Melaleuca styphelioides* exhibiting good activity against this fungus [120]. These same oils were evaluated for their antiviral activity in African green monkey kidney cells through the plaque reduction assay in the herpes simplex virus type 1 [121]. Other plants that have good antifungal activity are *M. piperita*, *Brassica nigra*, *Angelica archangelica*, *Cymbopogon nardus, Skimmia laureola, Artemisia sieberi*, and *Cuminum cyminum* [122,123,124,125,126,127]

Regarding antioxidant activity, the essential oil of the seeds of *Nigella sativa L*. showed considerable activity in the elimination of hydroxyl radicals. The essential oil of *M. armillaris* has marked antioxidant potential, changing the parameters of superoxide dismutase, and improving the concentrations of vitamin E and vitamin C [128]. However, there have been promising insect repellency/toxicity results from the essential oils of *Nepeta parnassica*, in *Culex pipiens molestus* [129]. Geranial, neral, geraniol, nerol, and trans-anethole are well established to stimulate the estrogenic response, and citrus (a combination of geraniol, nerol, and eugenol) is effective in replacing [^3^H] 17β-estradiol at the oestrogen receptors in recombinant yeast cells [130,131].

Cancer is a growing health problem worldwide and is the second leading cause of death. Essential oil constituents play an important role in cancer prevention and treatment as alternatives to synthetic drugs. Mechanisms such antioxidant, antimutagenic, and antiproliferative properties, enhancement of immune function and surveillance, enzyme induction and enhancing detoxification, modulation of multidrug resistance, and the synergistic mechanism of EO constituents are accountable for their chemo-preventive properties [132]. It has been reported that mitochondrial damage and apoptosis/necrosis in the yeast *Saccharomyces cerevisiae* were reduced by essential oils [133]. Recently, some studies demonstrated that certain EOs exhibited antimutagenicity towards mutations caused by UV light [8]. Jaganathan et al. reported that the active constituent eugenol from *Syzygium aromaticum* (cloves), nutmeg, basil, cinnamon, and bay leaves showed antiproliferative activity against several cancer cell lines in animal models [134]. 

In addition to medicinal and pharmacological applications, essential oils are used in perfumes, cosmetics, hygiene products, disinfectants, repellents, candles, phytochemicals, preservatives, and food additives. 

#### 2.3.3. Food Applications

In food, cosmetics, and personal care products, EOs are used as a natural aroma due to their chemical properties. In the food industry, EOs are being used as a food preservative because one of the main concerns is the preservation of food to prolong its useful life, ensuring safety and quality [11]. An expiration date is defined as the period of time during which the food product will remain safe. This ensures the maintenance of sensory, chemical, physical, microbiological, and functional characteristics. For example, spices can be encapsulated to extend their shelf life, maintain their properties, and inhibit reactions with other compounds [135]. Cinnamaldehyde, the aromatic agent present in cinnamon, has antimicrobial properties, and when encapsulated can slow the growth of yeasts in bakery products. Thus, the use of cinnamon in encapsulated form allows the product to be flavoured without interfering with the leavening process [136].

As the unpleasant taste and instability limit the application of EOs, the use of these encapsulated compounds can allow their application for several purposes. One of them is the intensification of the flavour of food products, where capsules can be used that release the product quickly when introduced into the mouth [20].

The packaging has the function of delaying deterioration, maintaining the quality and safety of packaged foods. For the packaging material to be satisfactory, it must be inert and scratch resistant, and must not allow molecular transfer to or from the packaging materials. Active packaging technologies extend the shelf life and control the quality of food products, decreasing microbial, biochemical, and enzymatic reactions through different strategies, such as adding chemical additives/preservatives, removing oxygen, controlling humidity and/or temperature, or a combination of these [137]. Oregano oil contains a high amount of carvacrol and is considered one of the most active plant extracts against pathogens due to its antimicrobial activity. Therefore, it has been used to preserve a variety of foods such as pizza, fresh beef [138], and cheddar cheese [139]. For the same purpose, limonene is reported for the preservation of strawberries [140], rosemary in chicken breast cuts [141], and cinnamon in pastries [142].

#### 2.3.4. Cosmetic and Cleaning Applications

In the detergent and cosmetics industry, microcapsules of essential oils are used in many products such as perfumes, creams, and deodorants where the controlled release of EOs is essential, increasing the duration of fragrance and the properties of the EOs [45]. Aroma ingredients such as patchouli (*Pogostemoncablin*), citronella (*Cymbopogon winterianus*), sandalwood (*Santalum álbum*), bergamot (*Citrusaurantium*), rosemary (*Rosmarinus officinalus*), mint (*Mentha piperita*), and vetiver (*Chrysopogon zizanioides*) are frequently used [4]. Regarding the EOs from flowers, *Lavandula officinalis*, rose, jasmine, tuberose, narcissus, and gardenia are those most commonly exploited for cosmetic applications [143]. Products such as detergents, soaps, shampoos, and softeners are largely produced using these natural compounds.

Over the years, EOs have also been used against nosocomial infections, as a cleaning liquid for disinfecting equipment and medical surfaces [9], or as an aerosol in operating rooms and waiting rooms to limit contamination [10].

#### 2.3.5. Agrochemical Applications

The loss of quality of agricultural products is caused by the presence of insect pests. The presence of these pests leads to reduced quality, low yield, and economic losses. Furthermore, human and animal health is compromised due to the production of carcinogenic secondary metabolites. To overcome this problem, chemical insecticides were used to excess. Despite being highly efficient, their overuse caused physiological resistance in several insect species and irreversible damage to the environment. Essential oils have emerged as a natural plant alternative to protect agricultural products from pests [144]. The use of EOs has intensified, mainly in gardens and homes, for pest control (Table 5), being important due to their toxic (pesticide) effect. EOs can be inhaled, ingested, or absorbed through the skin of insects. Monoterpenoids are an important group of chemical compounds in essential oils that interfere with the octopaminergic system of insects, which represent a target for insect control. As vertebrates do not have octopamine receptors, most chemicals in EOs are relatively safe to use. The special regulatory status together with the availability of essential oils has made the commercialisation of EO-based pesticides possible. Microencapsulation technology is used to produce these natural pesticides in order to mimic chemical compartmentalisation in plants, by protecting essential oils from degradation [145].

#### 2.3.6. Textile Applications

Essential oils are used in medical and technical fabrics. The technique used in industrial processes is encapsulation, which is used to give finishes and properties to textiles that were not possible or economical. The main application for encapsulation is durable fragrances and skin softeners, and other applications may include insect repellents, dye, vitamins, microbial agents, and phase-change materials, and medical applications, such as antibiotics, hormones, and other medications. 

The functionalisation of textiles with EOs with anti-mosquito repellent properties is a revolutionary way to protect human beings from insect bites and, thus, from many diseases such as malaria and dengue [146]. Plants, whose OEs have been reported to have repellent properties, include citronella, cedar, geranium, pine, cinnamon, basil, thyme, garlic, and mint. Khanna et al. performed the synthesis of a modified cyclodextrin host (β-CD CA) for inclusion complexation with the essential oils of cedarwood, clove, eucalyptus, peppermint, lavender, and jasmine for the assessment of repellent efficacy against *Anopheles Stephensi* in cotton. It was concluded that jasmine EO is the weakest against mosquitoes, as it worked as an attractant simulating flower nectar. Eucalyptus and clove are the feeding deterrents. On the other hand, lavender and peppermint are potential mosquito repellents, and cedarwood is an effective mosquito killer [147]. Soroh et al. reported that textiles treated with the Litsea and lemon EO microemulsion showed potential mosquito-repellent properties [148]. In general, citronella remains the most promising as an insect repellent and, therefore, is the most-incorporated EO in tissue functionalisation for this purpose. Specos et al. demonstrated citronella essential oil’s mosquito-repellent action, especially against *Aedes aegypti* [148]. Microcapsules with citronella are commonly incorporated into matrices such as cotton and polyester [149]. Another report determined that bio-based citronella oil has a better insect repellent effect than synthetic agents. Sariisik et al. concluded that, after washing, the insect repellent activity of the printing and coating method was increased, and the fabrics still showed repellency after five washing cycles [150]. 

## 3. Microencapsulation

Microencapsulation is the protection of small solid, liquid, or gaseous particles through a coating system (1–1000 mm) [151]. The encapsulated material is called the core and the material that forms the coating of particle is the wall or encapsulating agent [152]. Wall material can be a natural, synthetic, or semi-synthetic polymeric coating. In this technology, microparticles are formed, which can be classified in relation to their size and morphology, according to the encapsulating agent and microencapsulation method used [153]. 

Microparticles can be distinguished according to their form: they are classified as a reservoir-type system, ‘microcapsules’, when the core (encapsulated material) is concentrated in the central region, coated by a continuous wall material (encapsulating agent); or a monolithic system, ‘microspheres’, when the active agent (core) is dispersed in a matrix system (Figure 6). In general, the main difference is that in microspheres, part of the encapsulated material is exposed on the surface of the microparticle [154].

The physicochemical characteristics of the microcapsule are defined by the encapsulating agent and the active agent. The wall material must form a cohesive film that bonds with the encapsulated material [155]. Several materials can be used for the coating, with proteins, carbohydrates, and lipids being frequently used. Furthermore, the materials must be chemically compatible and the encapsulating agent chemically inert, so as not to react with the core [156].

Microencapsulation technologies achieve several objectives (Figure 7) and they are particularly used to protect the core active agent’s sensitivity to oxygen, light, and moisture, or to prevent interaction with other compounds. However, the most important reason for encapsulating an active agent is to obtain a controlled release [157].

The process of defining a microencapsulation system is mainly dependent on the purpose of the microcapsules. Characteristics such as shape, size, permeability, biodegradability, or biocompatibility are defined depending on the application of this material. Other physical and mechanical properties of the microcapsule, such as strength and flexibility, must also be defined [158].

One of the great advantages of microencapsulation is the mechanism of the controlled, sustained, or targeted release of the active agent. This release can occur at a certain defined time or not, through a mechanism of diffusion through or rupture of the wall. The release can be activated through temperature variations, solubility, pH changes, or even the biodegradability of the wall material [159].

Depending on the nature of the interaction of the encapsulating and encapsulated material, microencapsulation methods can be distinguished as chemical, physicochemical, and mechanical (Figure 8) [160]. In general, a microencapsulation method must be fast, easy, reproducible, and easily scalable for industry. The most-used microencapsulation methods are spray drying and coacervation, and these approaches will be mentioned in more detail below.

### 3.1. Emulsification

Emulsification is a fundamental step in oil microencapsulation, being used in a wide variety of food and pharmaceutical products. It is applied for the encapsulation of bioactive substances in aqueous solutions, which can be used directly in liquid or dried (spray-or freeze-drying) to form powders.

An emulsion consists of at least two immiscible liquids, with one of the liquids being dispersed as small spherical drops in the other. As can be seen in Figure 9, there are four systems, consisting of:Oil-in-water emulsion (O/W);Water-in-oil emulsion (W/O);Oil-in-water-in-oil emulsion (O/W/O);Water-in-oil-in-water emulsion (W/O/W).

In these systems, the droplet diameters can vary from 0.1 to 100 μm [161] and have been extensively revised by scientists [162]. The O/W emulsion consists of small oil droplets that are dispersed in an aqueous medium, being the droplets wrapped in a thin interfacial layer. Its advantages are the ease of preparation and low cost, with some disadvantages such as physical instability and limited control [162]. Through modifications of the emulsifiers, features can be added, such as the use of Maillard reaction products. These products can increase encapsulation efficiency and are able to protect the microencapsulation oil and other oils from oxidation [163].

A straightforward method for obtaining small droplets with a stratum size distribution is the evaporation/extraction of the emulsifying substance. This method is used in the preparation of biodegradable and non-biodegradable polymeric microparticles and in the microencapsulation of a wide variety of liquid and solid materials [164]. However, it is an expensive method with a low encapsulation efficiency, leading to residual solvent amounts [165].

### 3.2. Coacervation

Coacervation is one of the most widely used microencapsulation techniques. The technique is based on oppositely charged polyelectrolyte polymers that interact and form a wall covering the active agent. The coacervation process can be classified as simple and complex if one or two (or more) polymers are used, respectively. Generally, this technique is defined by the separation of two liquid phases in a colloidal solution, where one phase is rich in polymer (coacervated phase) and the other phase does not contain polymer (equilibrium phase) [46].

Complex coacervation involves the interaction of two oppositely charged colloids, where the neutralisation of charges induces a phase separation. A polysaccharide and a protein are usually used as the different polymers. Wall material systems that are most widely investigated include gelatin/gum arabic, gelatin/alginate, gelatin/glutaraldehyde, gelatin/chitosan and gelatin/carboxymethyl cellulose [166].

In the process of the microencapsulation of hydrophobic materials (Figure 10), the emulsification of the encapsulated agent in an aqueous solution containing two different polymers occurs, usually at a temperature and pH above the gel and isoelectric point of the protein. Then, the separation into two liquid phases (polymer-rich phase and aqueous phase) follows, which results from the electrostatic interaction of the polymers. Subsequently, a microcapsule wall is formed as the deposition of the polymer-rich phase occurs around the hydrophobic particles of the active agent, due to controlled cooling below the gelation temperature. Finally, the microcapsule walls harden through the addition of a crosslinking agent [167].

Simple coacervation has advantages over complex coacervation in terms of the associated cost, as cheap inorganic salts are used to induce the separation phase, while expensive hydrocolloids are applied in the complex method. Furthermore, complex coacervation is more sensitive to small variations in pH. However, compared to other microencapsulation methods, complex coacervation is a simple, scalable, inexpensive, reproducible, and solvent-free method, enabling its industrial use [166].

### 3.3. In Situ Polymerisation

In situ polymerisation (Figure 11) is based on the formation of a wall through the addition of a reagent inside or outside the core material [168], becoming one of the most-used methods in the preparation of microcapsules and functional fibres. Polymerisation takes place in the continuous phase and not on both sides of the interface between the core material and the continuous phase. Microcapsule formation occurs through an oil emulsion in a solution of melamine–formaldehyde resin and a sonication process to emulsify the oil in the aqueous phase. Then, resin is added under agitation and the pH is adjusted, with the formation of shells, thus promoting the reaction of the melamine with the formaldehyde at the interface of the oil droplets. This type of microcapsule is used in fragrances, insect repellents, food packaging, and footwear. The microcapsules result in smooth surface morphologies and are able to preserve the encapsulated scented oils for a sufficient period of time. They also have good thermal and controlled release properties [168,169].

Using a polymer as a microcapsule wrapper is considered a good addition due to its high strength and stability [170]. On the other hand, using a copolymer to prepare microcapsules with a low molecular weight of formaldehyde–melamine avoids the toxicity of formaldehyde [171].

In situ polymerisation is a method of rapid and easy expansion [172] and, at the same time, provides high encapsulation efficiency. However, the polymerisation reaction is difficult to control [173] and requires a large amount of solvent, making the monomers non-biodegradable and/or non-biocompatible [174].

### 3.4. Spray Drying

Spray drying is the most-used technology in the microencapsulation of essential oils. It is mainly used on an industrial scale, as it allows simple, reproducible, continuous, and low-cost production. Being used more frequently in the food industry, this process is also utilised in the cosmetics, pesticides, and pharmaceutical industries [175]. This technique allows encapsulated and powdered Eos to be obtained due to the ability to dry them in just one operation. In this process, the atomisation of emulsions occurs in a drying chamber with relatively high temperatures, where the evaporation of the solvent takes place and, consequently, microcapsules are formed [176]. 

The spray-drying technique involves four steps (Figure 12), where the preparation of dispersion first occurs, i.e., the wall materials are dissolved in water with agitation and controlled temperature. Still in the same step, the addition of the EOs follows and, if necessary, the emulsifier can be added. Afterwards, the dispersion is homogenised to be injected into the equipment through an atomising nozzle, where small droplets are formed. In the third step, emulsion atomisation occurs, where the formed droplets enter the drying chamber with a flow of hot air present. Finally, the dehydration of the atomised microparticles is done through the evaporation of the solvent, which dries the microparticles, which can them be recovered in the form of powder in a collector or filter [177]. 

The main limitations of this technique are related to the wall material, which must have good water solubility, and to the number of encapsulating agents available. In addition, some materials may be sensitive to the high temperatures introduced in the atomisation process. In addition, the production of microcapsules in fine powder form can cause agglomeration and an additional process may be required [166].

### 3.5. Freeze Drying

Freeze drying, also known as lyophilisation, is a simple process (Figure 13) that is used to dehydrate most materials sensitive to heat and aromas such as oils. Sublimation is the major principle involved in this drying process, where water passes directly from a solid state to a vapour state without passing through the liquid state. Before starting this process, the oil is dissolved in water and frozen [178]. Afterwards, the pressure is reduced and heat is added to allow the frozen water to sublimate the material directly from the solid phase to the gas phase. Freeze-dried materials appear to have the maximum retention of volatile compounds compared to spray drying, and this technique is used to microencapsulate some oils, with high yields [179]. This method helps to better preserve the EO content in many types of herbs and spices compared with other preservation techniques [180]. Lyophilisation is simple and easy to operate, showing that lyophilised samples are more resistant to oxidation and less efficient in microencapsulation [181]. The process disadvantages include high energy use, long processing time, and high production costs [182].

### 3.6. Supercritical Fluid (SCF) Technology 

Many pharmaceutical, cosmetic, and food industries use supercritical fluid technology (Figure 14) to form the microcapsules of essential oils due to their inherent advantages. The use of a wide variety of materials that produce controlled particle sizes and morphologies, the easy solvent removal, the non-degradation of the product, and being a non-toxic method are some of the many advantages of SCF technology. 

The methods used for supercritical fluids are the precipitation of gas anti-solvent, particles of saturated gas solutions, the extraction of fluid emulsions, and the rapid expansion of supercritical solutions [183,184]. The supercritical solvent impregnation process has proven to be successful in a wide variety of substances (essential oils, fragrances, active pharmaceutical compounds, and dyes) and matrices (wood, polymers, cotton, and contact lenses).

An alternative to spray drying (that degrades oils at high temperatures) is impregnation with supercritical solvent, as it is an ecological process where supercritical carbon dioxide is used as a green solvent.

### 3.7. Coaxial Electrospray System

The food, cosmetic, and pharmaceutical industries use a new technology to encapsulate oils, called coaxial electrospraying (Figure 15) [185,186]. This system is used in two phases, with external and internal solutions being sprayed coaxially and simultaneously through two feed channels separated by a nozzle.

In the electrospray process, the Taylor cone is composed of a core-shell structure that is formed at the top of the spray nozzle, ending up with the polymeric solution encapsulating the internal liquid. This method is distinguished by its ease and efficiency, and the maximum speed of the core material. The coaxial electrospray system provides a uniform size distribution, a high encapsulation efficiency, and an effective protection of bioactivity. However, the encapsulation efficiency and the stability of the microcapsules are affected by the wall materials [186]. Furthermore, controlling the process in coaxial electrospraying is difficult to some extent [187].

In experimental terms, the reported work on coaxial electrospray is based on individual laboratory experiments, consisting of specific combinations of materials and empirical process parameters. The fabrication of polymeric microparticles and nanoparticles is hampered by the lack of standard protocols. Regarding the collection of particles, the methodology cannot facilitate the hardening of the shell or maintain the morphology of particle, or even prevent its aggregation. On the theoretical side, many existing process models are empirical or semi-quantitatively empirical. The simulated results are not enough for the quantitative control of the process, as numerical simulations, such as computational fluid dynamics modelling, have been used to simulate the formation of the liquid cone and atomisation in a single axial electrospray process [188]. In summary, more experimental and theoretical study is needed to better understand the physical nature of coaxial electrospray and to provide quantitative guidance for process control.

### 3.8. Fluidized Bed Coating

Fluidised bed coating is one of the most efficient coating methods, in which the ingredients can be mixed, granulated, and dried in the same container. Consequently, the handling and processing time of the material is reduced. This approach was recently used to encapsulate fish oil by spraying and coating it (Figure 16) [189]. Fluidised bed coating is carried out by suspending the solid particles of the core material by an air stream under controlled temperature and humidity and then sprayed, building, over time, a thin layer on the surface of the suspended particles. This material must have an acceptable viscosity for atomisation, and the pumping should be able to form an appropriate film and be thermally stable [190].

There are several methods used in fluidised bed coating, including top spray, bottom spray, and tangential spray methods. In the top spray system, the coating solution is sprayed in the opposite direction with air in the fluid bed. The opposite flows lead to an increase in the efficiency of encapsulation and the prevention of agglomerates formation, achieving microcapsules with a size between 2 and 100 μm. The bottom spray, known as the Wurster system, uses a coating chamber that has a cylindrical steel nozzle (used to spray the coating material) and a cribriform bottom plate, coating small particles (100 μm). This multilayer coating procedure helps to reduce particle defects, although it is a time-consuming process. On the other hand, tangential spray consists of a coating chamber with a rotating bottom of the same diameter as the chamber. During the process, the drum is raised to create a space between the edge of the chamber and the drum. A tangential nozzle is placed above the rotating drum, where the coating material is released. Then, the particles move through the space into the spray zone and are finally encapsulated [191]. During this process, there are three mechanical forces, namely, centrifugal force, lifting force, and gravity.

The particles to be coated must be spherical and dense, and must have a narrow size distribution and perfect fluidity, with the non-spherical particles having the largest possible surface area and requiring more coating material.

This technique has a low operating cost and a high thermal efficiency process, allowing total temperature control. However, it can be time consuming, which becomes a disadvantage [173].

### 3.9. Layer-by-Layer Self-Assembly 

Layer-by-layer (LbL) is a relatively simple and promising technique for the encapsulation, stabilisation, storage, and release of several active compounds [192]. This method consists of alternating the adsorption of oppositely charged wall materials through many intermolecular interactions onto a charged substrate (Figure 17). The microcapsules have good chemical and mechanical stability through a formation mechanism constituted by irreversible electrostatic interactions that allow the adsorption of successive layers of polyelectrolytes [193]. The adsorption of the layers is normally carried out by immersing the suspension in alternate solutions of cationic and anionic polymers, with washing processes being carried out after the deposition of each layer [194]. This technique has significant advantages over other microencapsulation methods, because it allows the control of the permeability, morphology, composition, size, and wall thickness of the microcapsules by adjusting the number of layers and experimental conditions [195]. Controlling these parameters allows a better adaptation of the microcapsule to its functionality in the target application. However, most LbL systems have some restrictions in terms of biocompatibility [196].

## 4. Microencapsulation of Essential Oils

Microencapsulation is an alternative that can be utilised to overcome several limitations in the application of essential oils. This application is profoundly affected by the high volatility and chemically unstable nature of EOs [198]. In addition, EOs are compounds that can be easily degraded due to interactions with other chemical components and exposure to several factors such as light, temperature, and oxygen [166].

Essential oils can be “trapped” in microcapsules, which act as micro-reservoirs, ensuring excellent protection [199]. The encapsulation process, where small particles are enclosed in solid carriers to increase their protection, has the ability to reduce evaporation, promote easier handling, and control the release of essential oils during storage and application [199]. Furthermore, through microencapsulation, it is possible to change the appearance of EOs (which behave like a powder), without changing their structure and properties [177].

In EO microencapsulation, the first step is often to emulsify or disperse the essential oils in an aqueous solution of a wall material, which also acts as an emulsifier. This process happens because the EOs exist in liquid form at room temperature. Then, the resulting microcapsules must be dried under controlled conditions, so that the loss of the encapsulated material by volatilisation is reduced [177]. One of the areas that has also aroused interest in the microencapsulation of EOs is in the agrochemical industry. Yang et al. prepared and characterised microcapsules based on polyurea, containing essential oils as an active agent for possible applications in the controlled release of agrochemical compounds [200]. The microcapsules were synthesised by O/W emulsion interfacial polymerisation and the synthetic conditions that showed the best results were used to encapsulate four essential oils (lemongrass, lavender, sage, and thyme), capable of interfering with the seed germination and root elongation of some plants. In cases of pest control, biological pesticides must be more effective than synthetic pesticides. 

Bagle et al. reported success in encapsulating neem oil, an effective biological insecticide, in phenol formaldehyde (PF) microcapsules [201]. The microcapsules were obtained using an in situ polymerisation process in an O/W emulsion and their size was determined using a particle size analyser. Controlled release was monitored by measuring optical observations in the UV range. Figure 18 shows scanning electron microscopy (SEM) micrographs of PF microcapsules containing neem oil. It was possible to visualise that the PF microcapsules were spherical and globular, with diameters between 30 and 50 µm at 400–500 rpm. The microcapsules’ surface was considered quite smooth and can be useful regarding the protection and sustained release of the neem oil inside.

The chemical constitution of synthesised microcapsules was confirmed by Fourier-transform infrared spectroscopy (FTIR), and it was found to be a good thermal stability of MCs needed for the long-term preservation of the core, and it was concluded that neem oil can be better preserved in PF microcapsules.

The controlled release behaviour of PF microcapsules containing neem oil was studied and the experimental data are shown in Figure 19. A release of about 30% was observed after 6 h, confirmed by the decrease in absorbance over time. 

Like neem oil, other essential oils also have insecticidal properties, such as *Rosmarinus officinalis* and *Zataria multiflora* (Lamiaceae), that can be used as pesticides for stored-product pests. In the study carried out by Ahsaei et al., these oils were encapsulated in octenyl succinic anhydride (OSA) starch to test their insecticidal activity against *Tribolium confusum*. The microcapsules were obtained using an O/W emulsion and dried using the spray drying technique [202]. The solid formulations were characterised by particle size, encapsulation efficiency, and water activity. The release rate under storage conditions was measured over a period of 40 days, and the insecticidal activity against *T. confusum* was determined using specific bioassays. It was concluded that the encapsulation efficiency depends directly on the surfactant-to-oil ratio. Regarding the morphology of microcapsules loaded with OEs, SEM micrographs reveal the presence of oval and spherical microcapsules with irregular surfaces. The microcapsules appear to be devoid of cracks or fractures, which is an advantageous feature for protecting the oil. The results also showed an optimised release of pesticides from controlled release formulations, which maximises their biological activity for a longer time. 

The food sector is probably the sector where the microencapsulation of essential oils is most explored, with the encapsulation of flavours being one of the great interests of this industry. Flavours are necessary for some foods, to promote consumer satisfaction and the consumption of those products. Nevertheless, the flavour stability in foods has been a challenge for this sector in order to achieve quality and acceptability. 

For the encapsulation of a flavour, Fernandes et al. evaluated, by spray drying, the effects of the partial or total substitution of arabic gum with modified starch, maltodextrin, and inulin in the encapsulation of rosemary essential oil [203]. 

Regarding the characterisation of microcapsules, moisture content, wettability and solubility, density and apparent density, and oil retention was determined. From SEM observations (Figure 20), the authors found that there was no evidence of cracking in the particles using any of the encapsulating formulations, ensuring low gas permeability and thus better protecting the EO of rosemary. Differences were observed in the surface of each type of particle, showing that the particles have a spherical shape. It was concluded that the total substitution of arabic gum with modified starch or a mixture of modified starch and maltodextrin did not affect the efficiency of the encapsulation, increasing the possibility of developing new formulations of encapsulants. With the addition of inulin, the oil retention of particles decreased. However, the combination of modified starch and inulin was shown to be a viable substitute for arabic gum in foods.

A group of researchers compared the release properties of three different microcapsules, namely gelatin microcapsules loaded with holy basil essential oil (HBEO) (designated as UC), UC coated with aluminium carboxymethylcellulose (CC), and UC coated with aluminium compound carboxymethyl cellulose–beeswax (CB) [204]. To be applied as a feed additive, the HBEO was encapsulated in order to be a potential alternative to antibiotic growth promoters (AGP). However, its benefits depend on the available amount in the gastrointestinal tract.

The SEM technique was used to characterise the internal and external factors of the microcapsule surface morphology. According to Figure 21, UC microcapsules (Figure 21a) are almost spherical in shape and after coating, the CC (Figure 21b) and CB (Figure 21c) microcapsules are more spherical. Upon magnification of these micrographs, it was possible to verify that UC microcapsules have a spongy structure (Figure 21d) and that CC (Figure 21e) and CB (Figure 21) microcapsules are denser. When cut transversely, UC microcapsules seem to have a gelatinous morphology (Figure 21g), whereas the CC (Figure 21h) and CB microcapsules (Figure 21i) reveal a thicker and more compact outer coating layer with a honeycomb structure. This method of encapsulation demonstrated an effective process for improving HBEO efficacy for pathogen reduction in the distal region of the intestine.

Regarding food safety, the use of antimicrobial packaging materials offers the potential to retard the growth rate of spoilage microorganisms. The physical and antimicrobial properties of nanofibres manufactured for active packaging systems were studied by Munhuweyi et al. [205]. Microcapsules and active nanofibres derived from the precipitation of β-cyclodextrin (β-CD) with essential oils of cinnamon and oregano were developed and their antifungal activity in vitro against *Botrytis* sp. was examined. To induce microencapsulation, the solutions were subjected to co-precipitation. It was verified that cinnamon microcapsules have greater antimicrobial efficacy when compared to oregano. As food preservatives, this microencapsulation system could have promising applications in the development of active packaging systems.

Using the thermogravimetric analysis (TGA) technique, the initial weight loss for simple β-CD occurred at ~100 °C and the greatest weight loss at ~330 °C (Figure 22a). The degradation temperature of β-CD in the CIN/β-CD and OREG/β-CD complexes decreased from ~330 °C to ~270 °C (Figure 22b,c). Comparing the TGA curves, there is a difference between them, demonstrating the presence of chemical and guest molecule interaction in the complex.

Using the simple coacervation method, Leimann et al. encapsulated lemongrass, which is known for its broad spectrum antimicrobial activity [206]. Poly(vinyl alcohol) crosslinked with glutaraldehyde was used as the wall-forming polymer. The influence of the agitation rate and the fraction of oil volume on the microcapsule size distribution was evaluated. Sodium dodecyl sulphate (SDS) and poly(vinyl pyrrolidone) were tested to prevent the agglomeration of microcapsules during the process. The microcapsules did not show agglomeration when 0.03% by weight of SDS was used. The composition and antimicrobial properties of the encapsulated oil were determined, demonstrating that the microencapsulation process did not deteriorate the encapsulated essential oil. 

Cyclodextrins (CDs) are important supramolecular microcapsule hosts in foods and other fields, and the essential oil of *Laurus nobilis* (LEO) has natural antioxidant properties in food due to its main constituents being terpenic alcohols and phenols. For these reasons, Li et al. isolated LEO by microwave-assisted hydrodistillation [207]. The authors prepared chitosan (CS) microcapsules loaded with citrus essential oils (CEOs: D-limonene, linalool, a-terpinene, myrcene, and a-pinene) using six different emulsifiers (Tween 20, Tween 40, Tween 60, Tween 60/Tween 20/Span 80 1:1, Tween 20/sodium dodecyl benzene sulfonate (SDBS) 1:1, Span 80) through an emulsion gelation technique [208]. After preparing β-cyclodextrin (β-CD) microcapsules and their derivatives, several affecting factors were examined in detail. 

Figure 23 shows the total antioxidant activity of LEO. LEO caused Mo (VI) to be deoxidised to become Mo (V) through a mechanism of total antioxidant activity. Mo (V) exhibits maximum absorption at 695 nm and has a stronger antioxidant activity; the greater the concentration of Mo (V) solution, the greater the absorbency of the solution. With the increase in absorbance of the solutions, there was an increase in the concentrations of the sample, causing the antioxidant activity to increase significantly.

The microcapsules were analysed and the results indicate that the choice of emulsifier significantly affects the size and effectiveness of incorporating the microcapsules. 

Figure 24a presents the FTIR spectra observed in CS, CEOs, and four groups of microcapsules prepared with different emulsifiers. In the CEOs curve, the peak at 886 cm^−1^ corresponds to the absorption of limonene. The strong methylene/methyl band occurs at 1435 cm^−1^, and at 1646 cm^−1^, the C=O stretching vibration appears. Peaks corresponding to the asymmetric and symmetrical modes of the CH_2_ elongation vibration appear for Span 80 and Tween 60, and the new connections can be seen at 2922 cm^−1^. Through these results, it was possible to observe that the CEOs were incorporated in the microcapsules, showing benefits for inhibiting them from oxidation and volatilisation.

A second step in the characterisation of the microcapsules was the analysis of the crystallographic structure. Through X-ray diffraction (XRD) analysis (Figure 24b), it was possible to observe that CS exhibits a diffraction pattern with a broad band centred at 2θ 20°, thus indicating the existence of an amorphous structure. Comparing the CS with the microcapsule groups, the latter exhibit a significant reduction in this broad band. This reduction in intensity is due to the destruction of the CS structure, which can be attributed to a change in the arrangement of the molecules in the crystalline chain.

To develop a new use of functional EOs, Karimi Sani et al. studied the influence of process parameters on the characteristics of microencapsulated essential oil *Melissa officinalis* using whey protein isolate (WPI) and sodium caseinate (NaCS). The impacts of these variables were examined using the response surface methodology. Smaller particle sizes were obtained for higher amounts of WPI with the lowest level of applied sonication power. The results of the desirability function indicate that the maximum amount of WPI with an ultrasound power of 50 W led to the smallest particle size and the lowest zeta potential and turbidity. In this study, the ultrasonic technique showed potential in the use of milk proteins to produce microparticles with OEs. The obtained results showed that the microcapsules loaded with *Melissa officinalis* can preserve the bioactive compounds and induce flavour stability, enabling their use in food formulations and pharmaceutical products. 

Mehran et al. carried out a study of the microencapsulation of spearmint essential oil (SEO), using a mixture of inulin and arabic gum as wall material in order to be used in the food and pharmaceutical industry [209]. The technique used for the formation of the microcapsules was spray drying. The microcapsules were characterised for oil retention, encapsulation efficiency, hygroscopicity, and carbon content, having as ideal conditions 35% solid wall, 4% essential oil concentration, and 110 °C inlet temperature, with maximum retention of 91% of oil. To confirm that the SEO was encapsulated, this group of researchers used differential scanning calorimetry (DSC) and FTIR characterisation techniques.

The infrared spectra of pure SEO, pure matrix (containing inulin and arabic gum), and microcapsules are shown in Figure 25. In the SEO spectrum, the characteristic peaks at 801 cm^−1^ and 894 cm^−1^ are ascribed to =CH vibrations. The C-O-C elongation corresponds to the peak at 1109 cm^−1^ and the C=O elongation corresponds to the peak 1675 cm^−1^. In the matrix spectrum, a wide band at 3392 cm^−1^ is related to the hydroxylated group. In relation to the peak at 1030 cm^−1^, it can be associated with the strong absorption bands of the C-O-C elongation. In the microcapsule spectrum, it can be observed that it is quite similar to the matrix, and that the peaks related to the SEO disappear or are absent, which may be related to the overlap of the peaks of the matrix and SEO due to the low weight fraction of SEO in the total weight of the microcapsules. Through this spectrum, it was possible to verify the successful encapsulation of the SEO (peaks at 1673 cm^−1^ and 900 cm^−1^).

The double barrier release system is a method used for essential oils that have antifungal activity, even against drug-resistant fungi. However, there are some limitations due to the sensitivity to pH, temperature, and light. Adepu et al. encapsulated three essential oils (thymol, eugenol, and carvacrol) in a polylactic acid shell with high encapsulation efficiency to achieve their synergistic antifungal activity using the coacervation phase separation method. These were incorporated into bacterial cellulose (a nanofibre fibrous hydrogel) [210]. An antifungal test was performed on the *Candida albicans* fungus model (a cause of common oral and vaginal infections). Another test was carried out—a transvaginal drug release study in vitro—to compare the release of microcapsules like colloids and composites, where the latter exhibited a controlled release. Through several studies, such as the SEM technique, it was found that the average size and size distribution of the microcapsules depends on the concentration of the used polymer (poly(lactic acid)) and surfactant (poloxamer).

SEM images of BC loaded with microcapsules demonstrate a regular distribution and spherical shape, appearing to be well separated and stable in the stages of the preparation process (Figure 26). From the highest magnification image, it was observed that the microcapsules were anchored to the nanofibre matrix.

Repellent essential oils are becoming increasingly widespread due to their low toxicity and customer approval. Its application in textile materials has been widely developed. To optimise their application efficiency, it is important to develop long-lasting repellent textiles using OEs. Specos et al. obtained citronella-loaded gelatin microcapsules through the complex coacervation method, which were applied to cotton fabrics in order to study the repellent effectiveness of the obtained fabrics [70]. The release of citronella by the treated tissues was monitored and the repellent activity evaluated by exposing a human hand and arm covered with the treated tissues to *Aedes aegypti* mosquitoes.

It was found that the tissues treated with citronella microcapsules present greater and more lasting protection against insects in comparison to fabrics sprayed with an ethanol solution of essential oil. Repellent textiles were obtained by filling cotton fabrics with microcapsule sludge, using a conventional drying method. This methodology does not require additional investments for the textile finishing industries, which is a desirable factor in developing countries. Figure 27A shows the morphology of blackberry-type microcapsules in a fresh paste with diameters ranging from 25 to 100 µm, while Figure 27B shows SEM micrographs of spray-dried microcapsules revealing two types of structures, with small spherical units of less than 10 µm and clusters ranging from 25 to 100 µm. 

In 2016, Ribeiro et al. investigated the functionalisation of photocatalytic titanium dioxide nanoparticles on the surface of polymeric microcapsules as a way to control the release of citronella by solar radiation, thus obtaining a release of a repellent without mechanical intervention [211]. These authors used a modified hydrothermal sol-gel method to synthesise TiO_2_ nanoparticles. Through several characterisation techniques, these authors were able to observe the surface of the microcapsules and the release efficiency. Using in vitro biological assays with live mosquitoes, the controlled release repellence effect of these photocatalytic microcapsules was reinforced by the inhibition of these vectors. According to the results, it was shown that functionalising the microcapsules with photocatalytic nanoparticles on the surface, and then exposing them to ultraviolet radiation, effectively increased the emission of citronella into the air, repelling mosquitoes. Table 6 shows an overview of illustrative examples of EO microencapsulation oils, wall materials, and microencapsulation methods with industrial importance.

## 5. Conclusions

This review summarises different types of EO structures and describes their extraction and application methodology. In addition, different techniques for microencapsulating essential oils are described and some reports are presented to provide a basis for research and industrial development. 

As described in this paper, EOs are used in several applications in the pharmaceutical, cosmetic, agricultural, and food industries, as they are natural metabolites produced by plants with interesting properties. Furthermore, EOs are being explored as an alternative to synthetic products due to their ecological factors and the fact that their characteristics are different from the corresponding synthetic product. For example, synthesised oil may have the same odour as natural oils, but may not have the same therapeutic characteristics.

Currently, there is growing interest in the application of EO microencapsulation, making it an effective and important tool in the preparation of high-quality products, improving their chemical, oxidative, and thermal stability. Besides these advantages, the shelf life, biological activity, functional activity, controlled release, physicochemical properties, and general quality of oils can also be improved with microencapsulation technology.

Based on the scientific studies available and presented throughout this paper, it can be concluded that the microencapsulation of EOs is an emerging trend for industrial applications. However, this development has limitations, such as the low diversity of wall materials and their incompatibility with microencapsulation methods. Many of the encapsulating agents available present a high cost for production on an industrial scale. In future research, microencapsulation must also be directed to encapsulate a different mixture of oils by different techniques, in order to disguise the flavour of the oils and to improve safety, quality, and nutritional value.

## Figures and Tables

**Figure 1 polymers-14-01730-f001:**
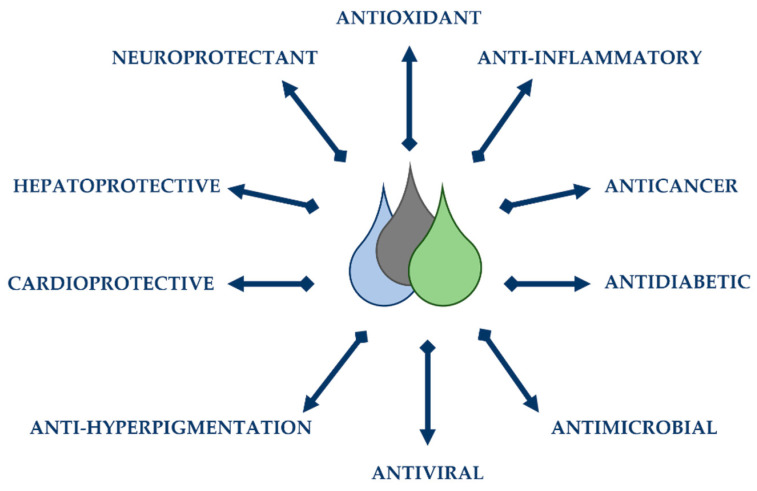
Biological activities of essential oils.

**Figure 2 polymers-14-01730-f002:**
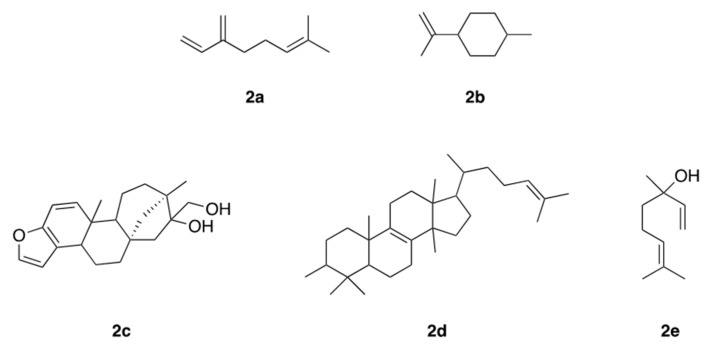
Structures of terpenes and terpenoids: acyclic monoterpenes (**2a**), cyclic monoterpenes (**2b**), diterpenes (**2c**), triterpenes (**2d**), and terpenoids (**2e**).

**Figure 3 polymers-14-01730-f003:**
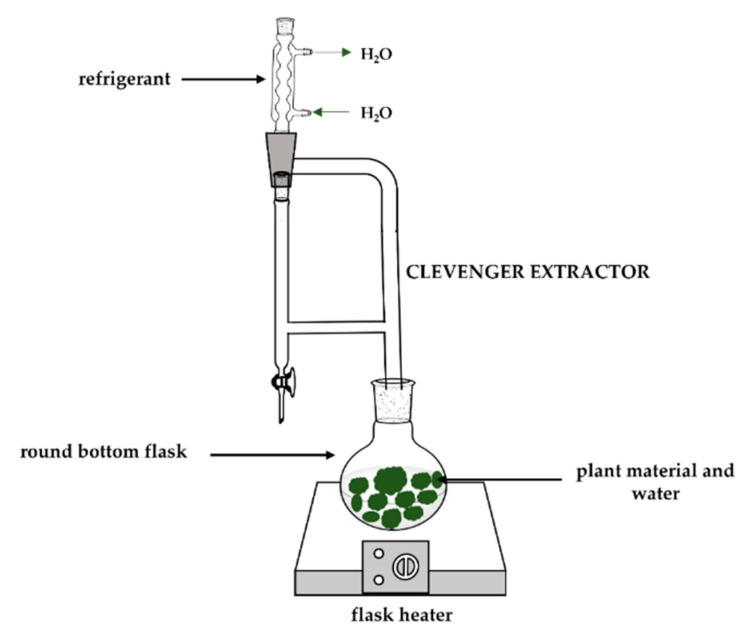
Schematic representation of hydrodistillation.

**Figure 4 polymers-14-01730-f004:**
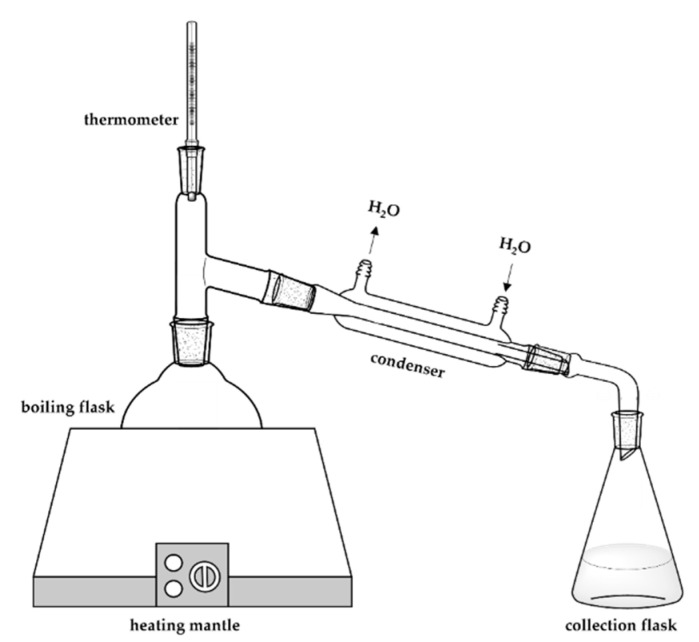
Experimental setup used in steam distillation.

**Figure 5 polymers-14-01730-f005:**
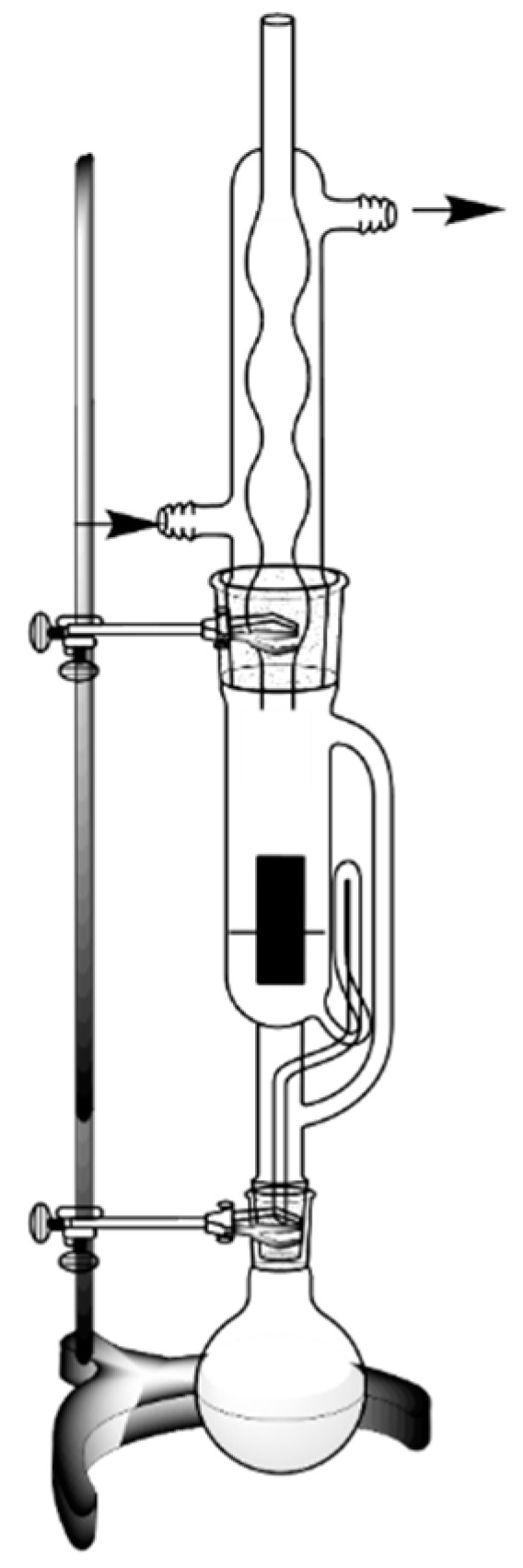
Schematic representation of organic solvent extraction using the Soxhlet method.

**Figure 6 polymers-14-01730-f006:**
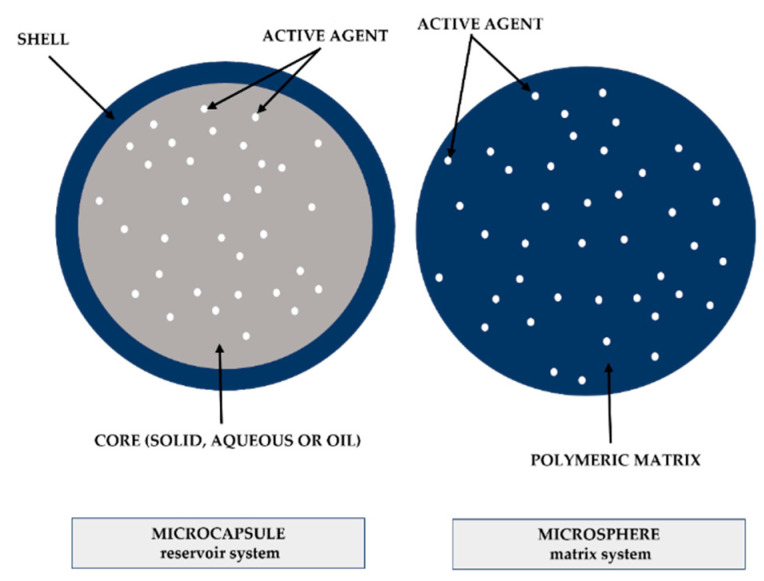
Schematic representation of a microcapsule and a microsphere.

**Figure 7 polymers-14-01730-f007:**
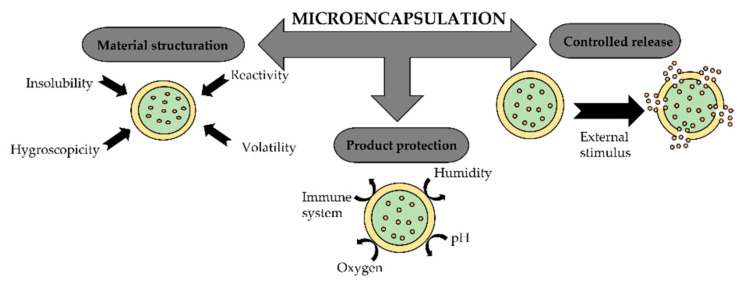
Objectives of microencapsulation.

**Figure 8 polymers-14-01730-f008:**
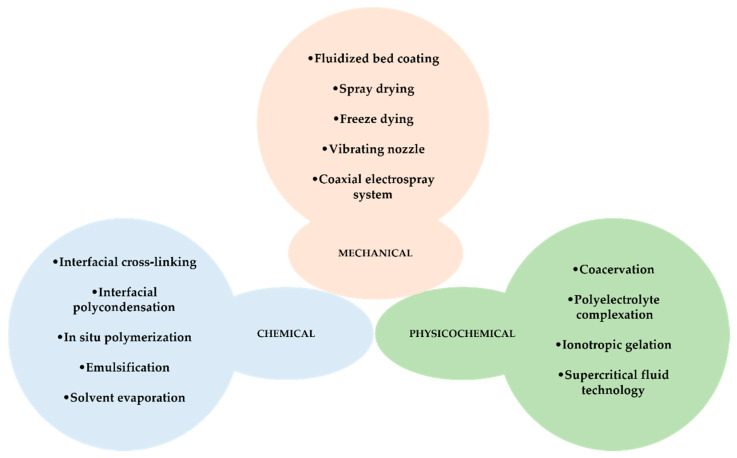
Main microencapsulation methods.

**Figure 9 polymers-14-01730-f009:**
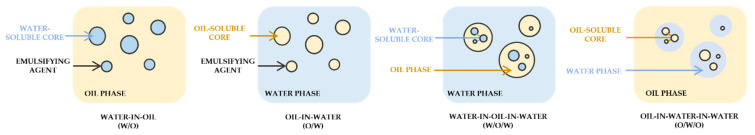
Illustration of emulsion systems.

**Figure 10 polymers-14-01730-f010:**

Schematic illustration of the coacervation method.

**Figure 11 polymers-14-01730-f011:**
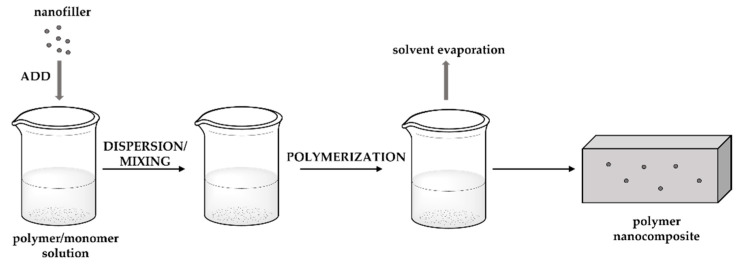
Schematic illustration in situ polymerisation method (adapted from [168]).

**Figure 12 polymers-14-01730-f012:**
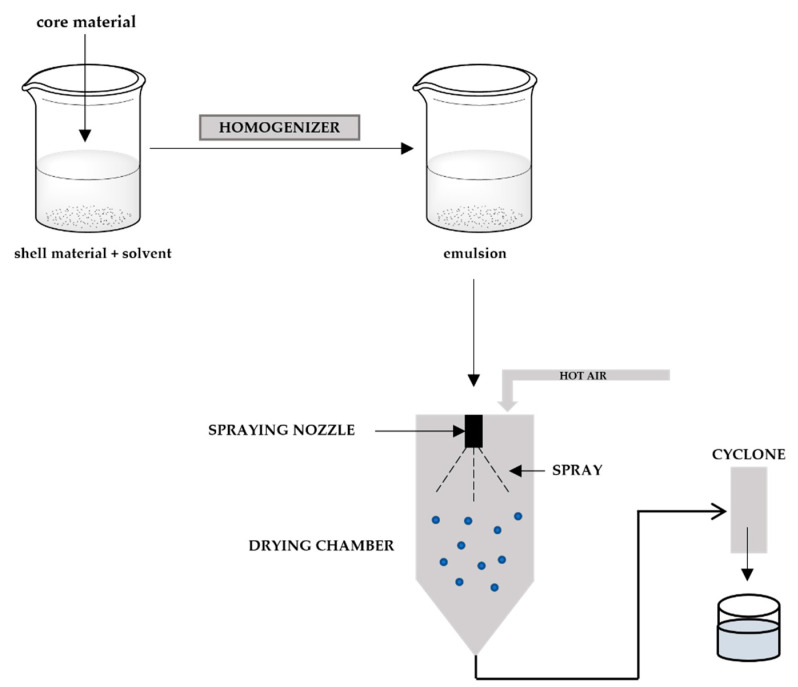
Schematic representation of spray drying.

**Figure 13 polymers-14-01730-f013:**
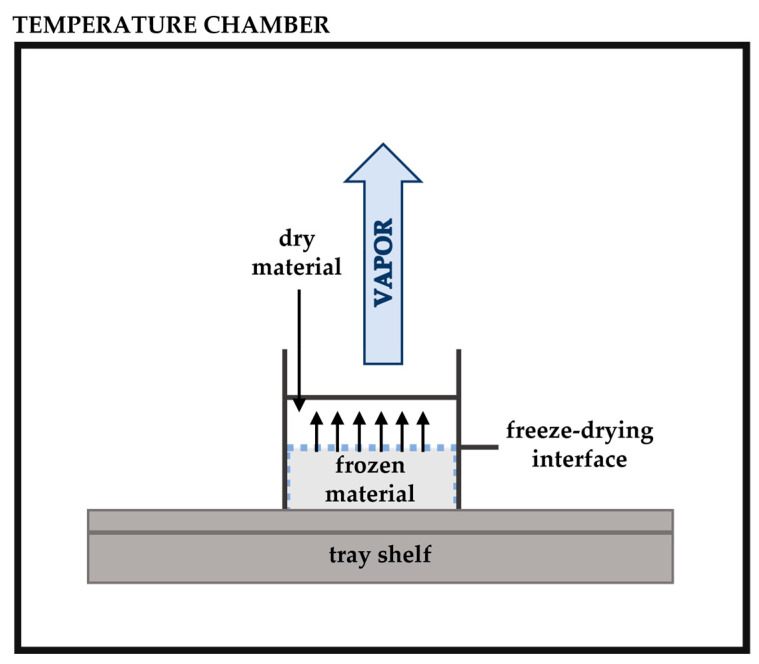
Schematic diagram of a freeze dryer (adapted from [163]).

**Figure 14 polymers-14-01730-f014:**
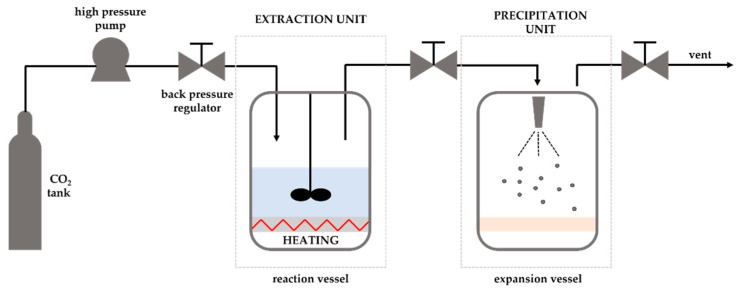
Flow chart of supercritical fluid technology.

**Figure 15 polymers-14-01730-f015:**
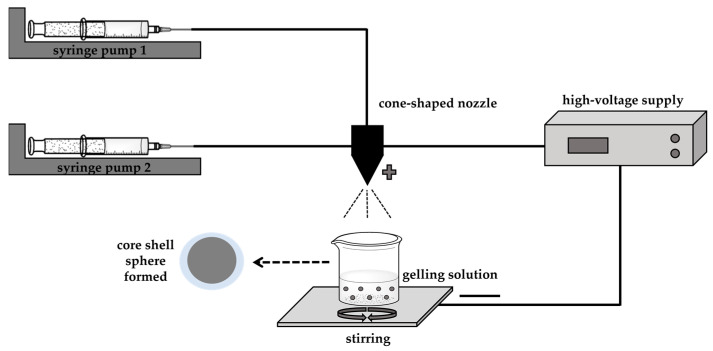
Schematic representation of microencapsulation process by coaxial electrospraying (adapted from [144]).

**Figure 16 polymers-14-01730-f016:**
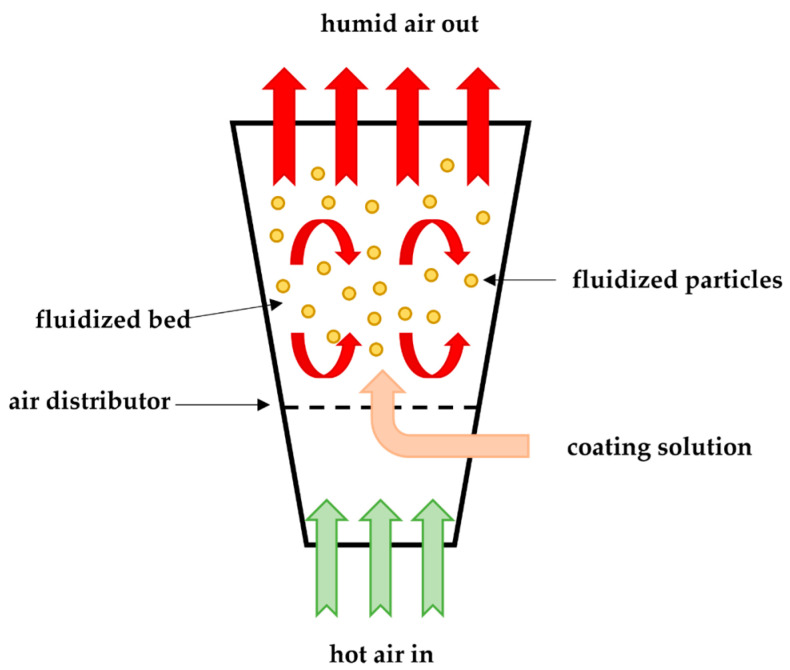
Schematic representation of bottom spray fluidised bed coating process (adapted from [166]).

**Figure 17 polymers-14-01730-f017:**
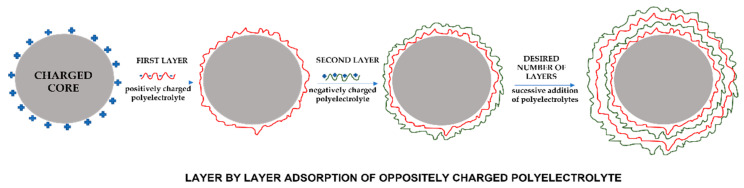
Layer-by-layer (LbL) self-assembly microcapsules (adapted from [197]).

**Figure 18 polymers-14-01730-f018:**
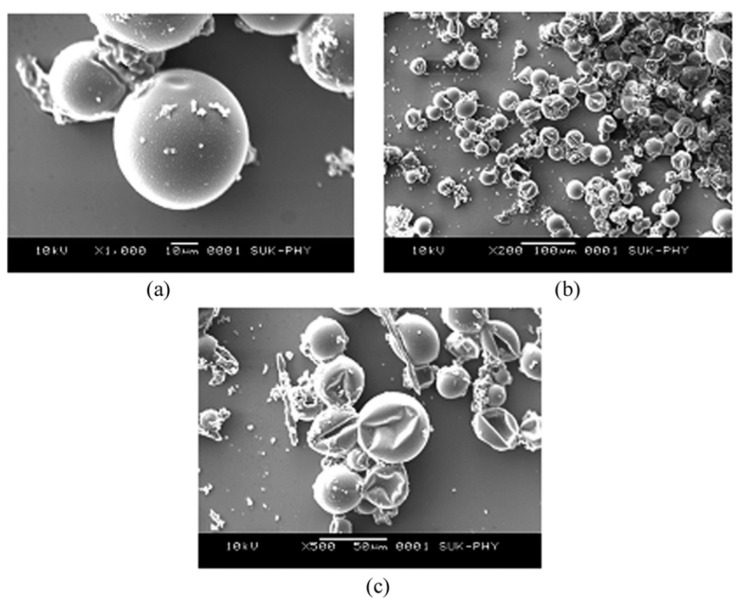
SEM micrographs (**a**–**c**) of phenol formaldehyde microcapsules containing neem oil [201].

**Figure 19 polymers-14-01730-f019:**
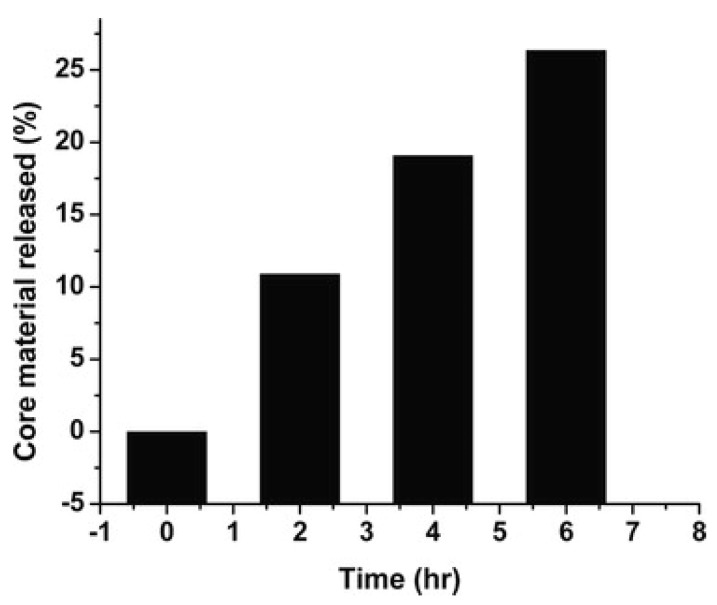
Controlled release of core material over time [201].

**Figure 20 polymers-14-01730-f020:**
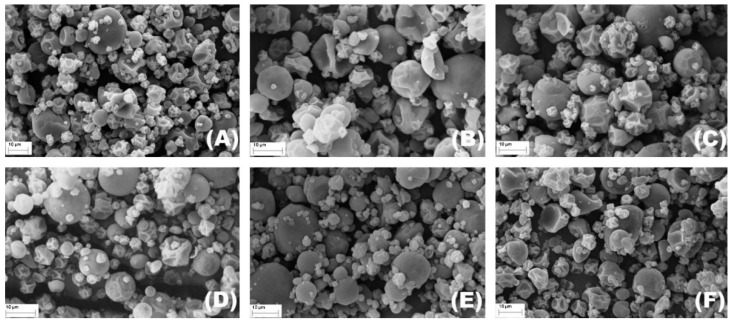
Scanning electron micrographs of the particles containing rosemary essential oil [203]. (**A**): arabic gum; (**B**): arabic gum/maltodextrin; (**C**): arabic gum/inulin; (**D**): starch; (**E**): modified starch/maltodextrin; (**F**): modified starch/inulin.

**Figure 21 polymers-14-01730-f021:**
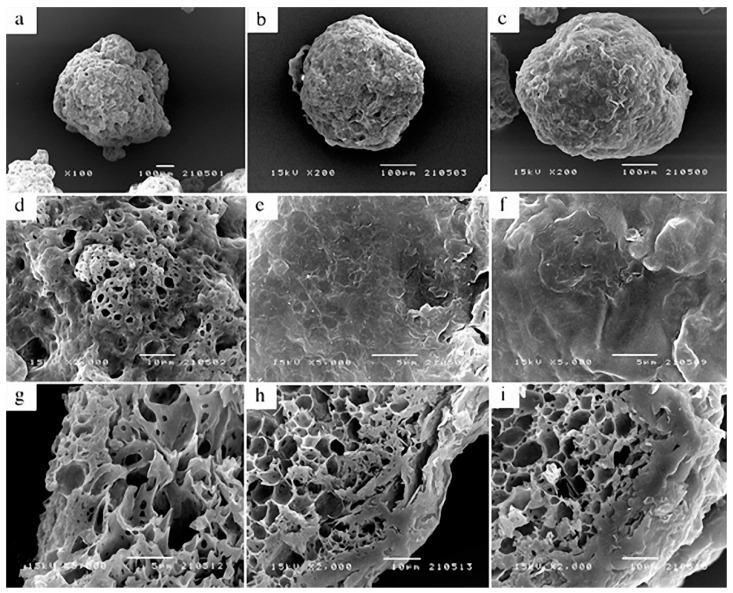
SEM micrographs of UC, CC, and CB gelatin-based microcapsules: (**a**) whole UC; (**b**) whole CC; (**c**) whole CB; (**d**) external surface of UC; (**e**) external surface of CC; (**f**) external surface of CB; (**g**) inner edge of UC; (**h**) inner edge of CC; (**i**) inner edge of CB [204].

**Figure 22 polymers-14-01730-f022:**
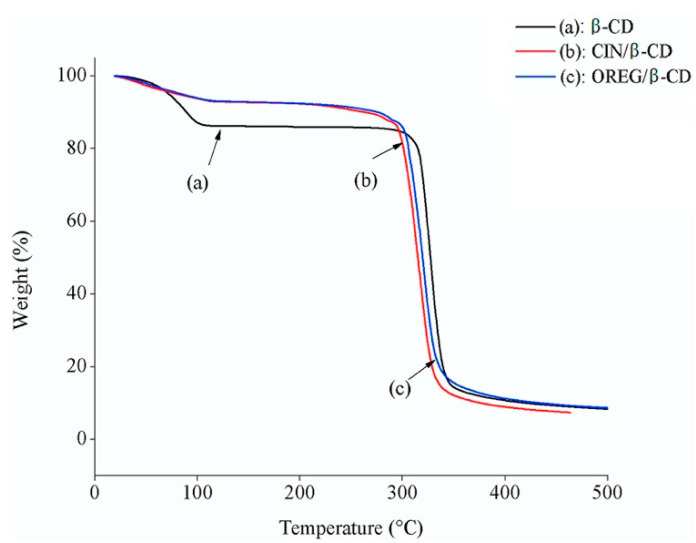
TGA curves of (**a**) plain β-cyclodextrin (β-CD), (**b**) microencapsulated cinnamon (CIN/β-CD), and (**c**) oregano (OREG/β-CD) [205].

**Figure 23 polymers-14-01730-f023:**
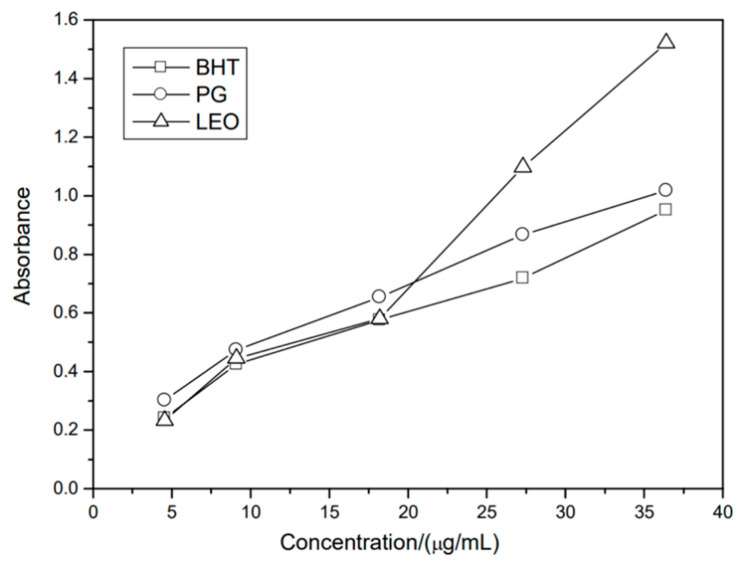
Absorbance of antioxidants in different concentrations of 2,6-ditert-butylphenol (BHT, square), propyl gallate (PG, circle) and *Laurus nobilis* essential oil (LEO, triangle) [207].

**Figure 24 polymers-14-01730-f024:**
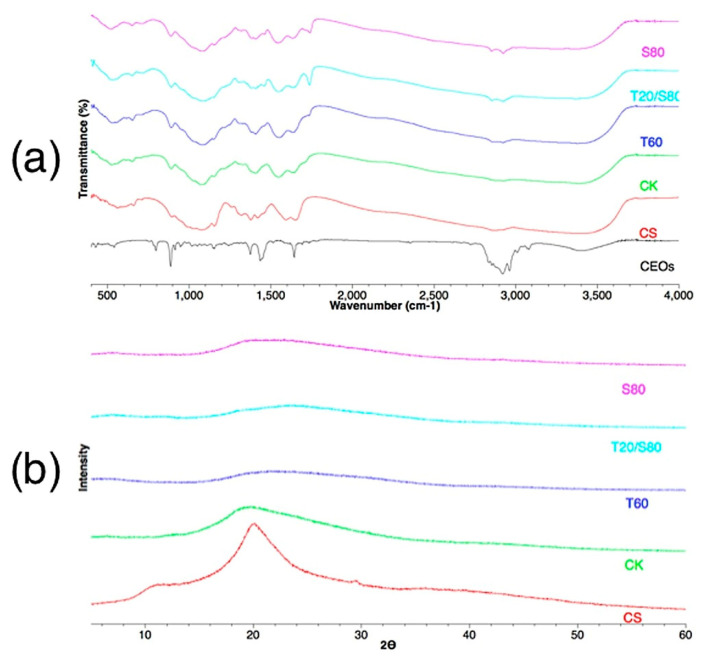
(**a**) FTIR spectroscopy, (**b**) X-ray diffraction of pure chitosan (CS), control group (CK), Tween 60, Tween 20/Span 80, and Span 80 [208].

**Figure 25 polymers-14-01730-f025:**
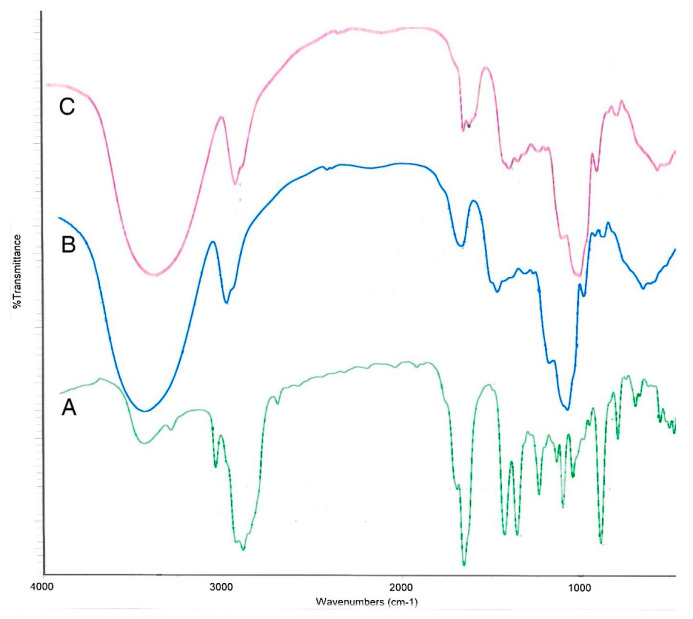
FTIR spectra of (**A**) pure SEO, (**B**) pure matrix, and (**C**) inulin and arabic gum-based microcapsules [209].

**Figure 26 polymers-14-01730-f026:**
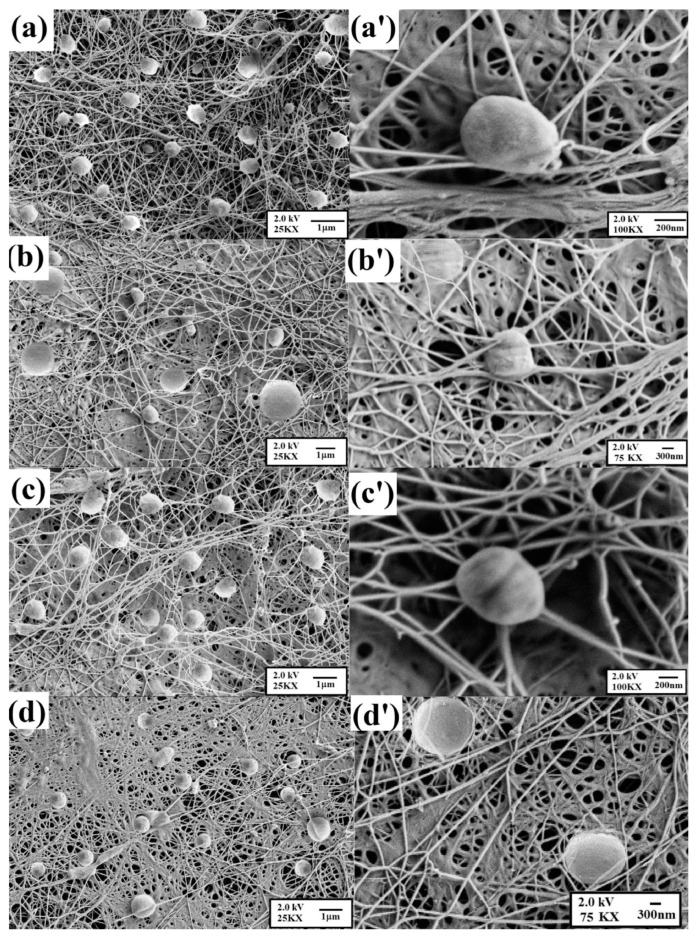
Low- and high-magnification SEM micrographs of (**a**,**a’**) BC-PLA_1.5_-Pol_5.0_, (**b**,**b’**) BC-PLA_1.5_-Pol_2.5_, (**c**,**c’**) BC-PLA_3.0_-Pol_5.0_, and (**d**,**d’**) BCPLA_3.0_-Pol_2.5_ [210].

**Figure 27 polymers-14-01730-f027:**
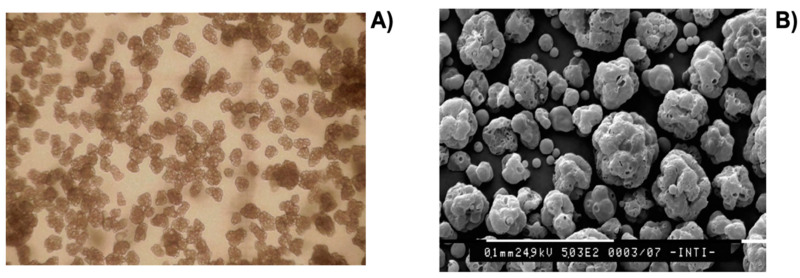
(**A**) Optical micrographs of gelatin microcapsules containing citronella essential oil (100× magnification) and (**B**) SEM microphotographs of spray-dried microcapsules containing citronella essential oil (500× magnification) [70].

**Table 2 polymers-14-01730-t002:** Advantages and disadvantages of each essential oil extraction method [46].

Type of Method	Method	Advantages	Disadvantages
**Conventional**	Hydrodistillation	- Versatile and simple; - Easy implementation; - Selectivity.	- Complete extraction is not possible; - High energy consumption; - Long extraction time.
	Steam distillation	- Extraction time and loss of polar molecules are reduced (compared with hydrodistillation).	- Longer extractions; - Present non-appreciable and higher cost compounds due to the long process time.
	Organic solvent extraction	- Simple, cheap, and reasonably efficient; - Appropriate for small scale.	- Time consuming; - High solvent consumption; - Does not allow agitation to speed up the process; - Organic solvents can cause chemical changes or toxic effects in final product.
	Cold pressing	- Simple and inexpensive; - Suitable for the production of citrus oils.	- Oil extraction is not complete; - Not feasible for low-oil samples.
**Innovative**	Supercritical fluid extraction	- Reduced time; - Low toxicity solvents; - Solvent-free extract.	- High cost of equipment, installation, and maintenance operations.
	Microwave-assisted extraction	- High reproducibility; - Simple manipulation; - Low solvent consumption; - Lower energy input; - Improved extraction yield.	- Filtration or centrifuging required at the end.
	Ultrasound-assisted extraction	- Simple and inexpensive (compared to SCFE and MAE); - Reduced extraction time; - Low solvent consumption; - Mass transfer intensification; - Improvement of solvent penetration.	- Difficult to scale up; - High power consumption.

**Table 3 polymers-14-01730-t003:** Essential oil plants, species, and main components, and their molecular structure and biological properties.

Plant	Species	EO Major Components	Chemical Structures of EOs Components	Some BiologicalProperties
Bay Laurel	*Laurus nobilis*	Cineol, pinene, linalool, terpineol acetate, methyleugenol	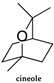 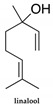 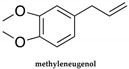	Antioxidant, antibacterial, antifungal, insect repellent
Citronella	*Cymbopogon nardus*	geraniol, citronellal, geranyl acetate, limonene, camphene	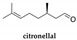 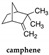 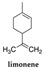	Antimicrobial, antifungal, antioxidant, antibacterial, insect and stimulate repellent
Hops	*Humulus lupulus*	Humulene, myrcene, caryophyllene, farnesene	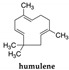 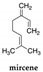 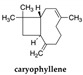	Antibacterial, antifungal, anti-cancer, repellent
Lemon Balm	*Melissa officinalis*	Geraniol, citral, citronellol, eugenol, linalyl acetate	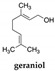 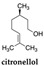 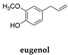	Antibacterial, antifungal, antimicrobial, sedative, antipyretic, antispasmodic, anti-hypertensive, anti-Alzheimer, antiseptic
Neem	*Azadirachta indica*	Azadiractin	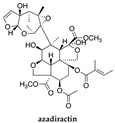	Insecticide, antiviral, antibacterial, antimicrobial
Pennyroyal	*Mentha Pulegium*	Pulegone, menthol, iso-mentone, octanol, piperitenone, trans-iso-pulegone	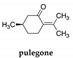 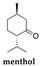 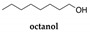	Insect repellent, antimicrobial, antioxidant, antibacterial, anti-tumour
Star Anise	*Illicium verum*	Trans-anethole	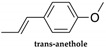	Antimicrobial, antioxidant, diuretic, anti-cancer potential, insect repellent
Valerian	*Valeriana officinalis*	Borneol, camphene, α and β-pinene, valeranone, valerenol	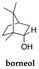 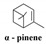 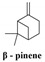 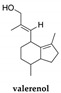	Antibacterial, antimicrobial, antifungal, antioxidative, sedative

**Table 4 polymers-14-01730-t004:** Some pharmacological actions of essential oils [114].

Condition	Essential Oil
Anxiety, agitation, stress, challenging behaviours	Angelica, labdanum, bergamot, sweet orange, palmarosa, lavender, basil, geranium, and valerian
End-of-life agitation	Lavender, sandalwood, and frankincense
Fatigue	Labdanum, grapefruit, coriander, citronella, black peppermint, gully gum, spearmint, geranium, Scots pine, clary, and ginger
Insomnia	Angelica, lemon, mandarin, sweet orange, lavender, lemon balm, myrtle, basil, sweet marjoram, and valerian
Mental exhaustion, burnout	Peppermint, basil, and everlasting
Memory loss	May Chang, peppermint, and rosemary
Pain management	Gully gum, lavender, German chamomile, sweet marjoram, rosemary, and ginger

**Table 5 polymers-14-01730-t005:** Pests/pesticides and their corresponding essential oil [145].

Pests	Essential Oil
Ants	Peppermint, mint
Aphids	Cedar, hyssop, peppermint, mint
Beetles	Peppermint, thyme
Caterpillars	Peppermint, mint
Mites	Lavender, lemongrass, sage, thyme
Fleas	Peppermint, lemon grass, mint, lavender
Flies	Lavender, mint, rosemary, sage
Mosquitoes	Patchouli, mint
Lice	Cedar, peppermint, mint
Moths	Cedar, hyssop, lavender, peppermint, mint
Slugs	Cedar, hyssop, pine
Snails	Cedar, pine, patchouli
Spiders	Peppermint, mint
Ticks	Lavender, lemongrass, sage, thyme

**Table 6 polymers-14-01730-t006:** Overview of essential oil microencapsulation, methods, wall materials, and industrial applications.

Microencapsulation Method	Wall Material(s)	Essential Oil/Source	Applications	Reference
Emulsification	Hydroxypropyl methyl cellulose/ chitosan/silica	Peppermint oil	Medical	[212]
	Polydopamine	Turpentine	Agrochemical	[213]
	β-cyclodextrin	Thyme	Food	[214]
	β-cyclodextrin/sugar beet pectin	Garlic	Food	[215]
Ionic Gelation	Cassava starch/poly(butylene adipate-co-terephthalate)	Oregano	Food	[216]
Simple Coacervation	Gelatin	Basil	Agrochemical	[217]
	Poly(vinyl alcohol)	Lemongrass	Food and pharmaceutical	[206]
Complex Coacervation	Gelatin/gum arabic	Citronella	Anti-mosquito textile	[149]
	Gelatin/sodium alginate	Citronella	Anti-mosquito textile	[218]
	Gelatin/gum arabic	Lavender	Cosmetics	[219]
	Chitosan/gum arabic/maltodextrin	Peppermint	Cosmetics	[220]
	Chitosan/k-carrageenan	*Pimenta dioica*	Food	[221]
	Gelatin/chia mucilage	Oregano	Food	[222]
	Mung bean protein isolate/apricot peel pectin	Rose	Food	[223]
	Sichuan pepper seed soluble dietary fibre/soybean protein isolate	Sichuan pepper	Food	[224]
	Whey protein isolate/arabic gum	Orange	Food	[225]
Interfacial Polymerization	Polyurea	Lemongrass, lavender, sage and thyme	Agrochemical	[200]
In situ polymerisation	Silicon dioxide/poly(melamine formaldehyde)	Cinnamon	Agrochemical	[226]
	Melamine/formaldehyde	Thyme	Food	[169]
	Melamine/formaldehyde	Lavandin and tea tree	Paints	[227]
Extrusion	Sodium alginate	Nutmeg	Pharmaceutical	[228]
	Sodium alginate	Rosemary	Agrochemical	[229]
Spray Drying	Acacia gum	Citronella	Cosmetics	[230]
	Palm trunk/ chitosan	Ginger	Food	[231]
	Maltodextrin	Citrus	Food	[232]
	Gum arabic/maltodextrin/sodium alginate/whey protein concentrate	Juniper berry	Food	[233]
	Whey protein isolate/maltodextrin/sodium alginate	Cinnamon	Food	[234]
	Hydroxypropyl methyl cellulose/maltodextrin	Oregano	Food	[235]
	Gelatin/arabic gum	Citronella	Textile	[70]
	Gum arabic/starch/maltodextrin/inulin	Rosemary	Food	[203]
	Gum arabic/modified starch	Black pepper	Food	[236]
	Gum arabic/maltodextrin/whey protein isolate	Basil	Food	[237]
Freeze-drying	Urushiol	Not mentioned	Medical and pharmaceutical	[238]
	β-cyclodextrin	*Litsea cubeba*	Cosmetics and pharmaceutical	[239]
	Maltodextrin/gelation	Lemongrass	Cosmetics, pharmaceutical and food	[240]
	Gum arabic/collagen hydrolysate	*Origanum* *onites* L.	Environmental	[241]
	Whey protein isolate/carboxymethylcellulose	Orange	Food	[242]
Supercritical Fluid Technology	Starch	Oregano	Food	[243]
	n-octenyl succinic/modified starches	Lavandin	Agrochemical	[244]
	Modified starch	Limonene	Food	[245]

## Data Availability

The raw/processed data required to reproduce these findings cannot be shared at this time due to technical or time limitations.

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
