# Peer review of "Microencapsulation of Essential Oils: A Review"

_polymers, 2022, doi:10.3390/polym14091730_

Round 1

Reviewer 1 Report

The authors have presented the review on the topic of microencapsulation of essential oils. The authors have properly prepared the paper. In the introduction part, they have explained the purposes of this review. In the second paragraph, they have defined the essential oils and described the methods of extraction of oils. The authors have also paid attention to the essential oil potential application. The third and fourth paragraphs are related to the problem of microencapsulation and microencapsulation of essential oils. 

The strength of this paper is: the timeliness of the paper topic, huge literature review (more than 200 references), enriching the review with numerous tables and figures.

However, the authors should some tables because they are unreadable (table 2, table 3).

Author Response

The authors are grateful for the reviewer's comments.

Regarding the formatting in tables 2 and 3, the authors feel that the present form aids visually to understand the advantages and disadvantages of each essential oil extraction method, as well as to distinguish the essential oils plants, species, main components and their molecular structure, and biological properties.

Reviewer 2 Report

The work presented in the review paper is very well-suited for publication in polymers. The authors include in this review the different microencapsulation strategies for general applications and for essential oils, some extraction methods for essential oils and also their applications.

  1. Overall, the review is very comprehensive, covering extraction methods of different types of oils, encapsulation, and applications. Tables are available in each section summarizing the content and providing readers with better comprehensiveness of the content being discussed. One point I would like to add is that in section 3 for general microencapsulation strategies. A major encapsulation technique has been left out. Layer-by-layer (LbL) self-assembly or polyelectrolytes or polymers is a very popular and common encapsulation strategy used for both hydrophilic and even hydrophobic cargoes such as oils. A brief discussion of the field would make the review quite complete. Here are some examples for you to consider citing (https://doi.org/10.1039/C4RA04750H and https://doi.org/10.1021/am401871u).
  2. Figures 2 and 3 are distorted in the pdf. Please check the figures, the conversion from word to pdf might have caused some distortion or maybe the figures need adjustment.

Author Response

The authors acknowledge the importance of the reviewer's comments. The English was revised.

The LBL technique has been introduced in the new section 3.9, as well as the recommended literature references.

Figures 2 and 3 graphical quality has been revised.